# Early stages of covalent organic framework formation imaged in operando

Christoph G. Gruber[1], Laura Frey[2], Roman Guntermann[2], Dana D. Medina[2✉] & Emiliano Cortés[1✉]

Covalent organic frameworks (COFs) are a functional material class able to harness, convert and store energy. However, after almost 20 years of research, there are no coherent prediction rules for their synthesis conditions. This is partly because of an incomplete picture of nucleation and growth at the early stages of formation. Here we use the optical technique interferometric scattering microscopy (iSCAT)[1–3] for in operando studies of COF polymerization and framework formation. We observe liquid–liquid phase separation, pointing to the existence of structured solvents in the form of surfactant-free (micro)emulsions in conventional COF synthesis. Our findings show that the role of solvents extends beyond solubility to being kinetic modulators by compartmentation of reactants and catalyst. Taking advantage of these observations, we develop a synthesis protocol for COFs using room temperature instead of elevated temperatures. This work connects framework synthesis with liquid phase diagrams and thereby enables an active design of the reaction environment, emphasizing that visualization of chemical reactions by means of light-scattering-based techniques can be a powerful approach for advancing rational materials synthesis.

The success story of ordered, porous molecular frameworks is based on the ability to rationally design their molecular building elements, which—in turn—enables control over their functionality. Particularly, 2D COFs[4] have shown intriguing properties as photocatalysts/electrocatalysts[5,6], proton conductors[7] or cathodes in batteries[8,9]. However, the quality of COF materials is highly dependent on their synthesis conditions. Here, in contrast to the molecular design, the design of reaction media, catalysts or reaction parameters such as pressure and temperature mainly relies on 'wisdom of crowd' methodologies[10–13].

To enable a rational design of their synthesis conditions, it is crucial to explain the underlying processes leading to COF formation. Obvious blind spots in their reaction landscape are the early stages during and after initiation of the polymerization reaction by a catalyst. Therefore, it is essential to monitor the reaction at a starting point of solvent mixing and at the initial stages of molecular interactions. Current methods struggle to access these early stages. The technique used would have to account for high temporal and spatial resolution in combination with high sensitivity to all matter present in the complex mixture—for example, crystalline, amorphous, porous/non-porous, liquid/solid phases—under realistic conditions (in operando).

Here we establish an optical technique, iSCAT[1–3], for holistic in operando studies on chemical reactions such as polymerization and formation of framework materials. iSCAT combines a sub-5-nm sensitivity at high speed (µs/ms) with spatial information and nanoscale localization precision (<10 nm)[14–16]. Furthermore, iSCAT offers label-free and universal sensitivity, as its detection principle relies on light scattering, which is inherent to all matter[17–19] (comparison of iSCAT with current techniques in Supplementary Information section 1).

To collect comprehensive insights into the early stages of COF formation, we chose the 2D imine COF termed TA-TAPB (TA, terephthalaldehyde;

TAPB, 1,3,5-tris(4-aminophenyl)benzene) as a model (Fig. 1a) representing imine-bond-connected COFs (more than 200 reported structures)[10,20] and the conventional solvothermal COF synthesis in ternary solvent systems[21]. Moreover, the reaction is carried out in the most often used solvent system (1,4-dioxane/mesitylene/aqueous acetic acid)[22].

## Operando imaging of imine COF formation

The typical COF materialization process can be described as a one-pot multistage synthesis (Fig. 1a,c). In common practice, every conventional solvothermal COF synthesis essentially starts at room temperature, at the point of mixing solvents, precursors and catalyst. Because we are aiming at collecting information at the very early stages of the process (milliseconds to minutes), we imaged the reaction with iSCAT at room temperature using 3 M acetic acid as catalyst (Fig. 1b). Notably, the formation of a COF material under the chosen experimental conditions has been verified (see discussion in Supplementary Information section 2).

Briefly, in iSCAT, the sample (that is, the reaction mixture) is located above a transparent coverglass and is illuminated with coherent light (here $\lambda$ = 785 nm), typically from below (Fig. 1b and Supplementary Fig. 15). Part of this incident light will be reflected at the coverglass–sample medium interface owing to a refractive index difference. Another part of the light will be scattered by the sample (probe region approximately 300 nm deep into the solution[18]). At the camera detector, the scattered light as well as the reflected light are collected. The iSCAT signal arises from the constructive or destructive interference of the scattered light with the reflected light (for the full methodology, see Methods and Supplementary Information section 3).

[1]Nanoinstitute Munich and Center for NanoScience (CeNS), Faculty of Physics, Ludwig-Maximilians-Universität München, Munich, Germany. [2]Department of Chemistry and Center for NanoScience (CeNS), Ludwig-Maximilians-Universität München, Munich, Germany. ✉e-mail: dana.medina@cup.lmu.de; emiliano.cortes@lmu.de

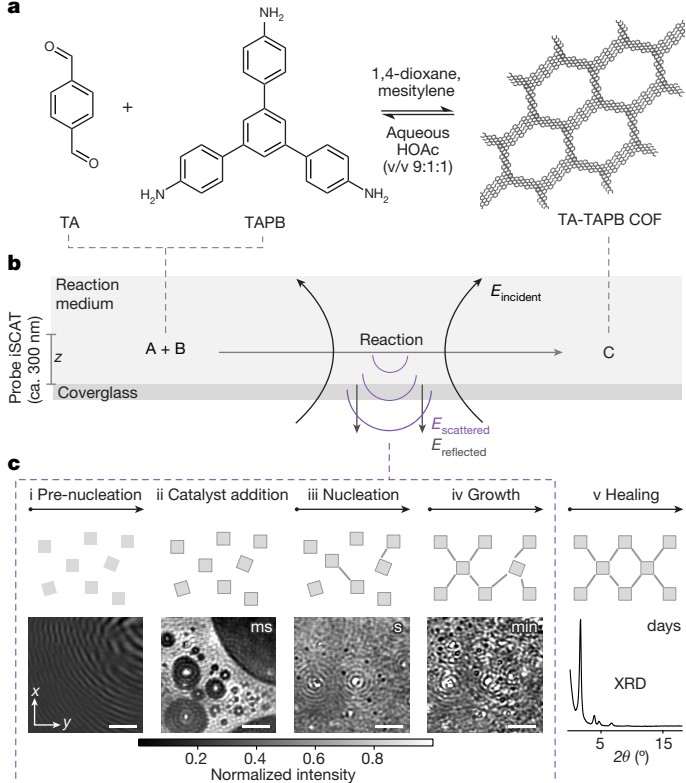

**Fig. 1 | iSCAT as operando tool to obtain holistic insight into the mechanism of COF formation. a**, Reaction scheme of the TA-TAPB COF formation. The reaction is carried out in a ternary solvent system of 1,4-dioxane, mesitylene and the catalyst mixture, water/HOAc (v/v 9:1:1). **b**, Working principle scheme of iSCAT. The incident light is partly reflected at the interface between the coverglass and the reaction medium. The reflected light interferes with scattered light created by the entities and processes in the reaction mixture and produces the iSCAT signal. The iSCAT image shows the surface of the coverglass and a probe region of around 300 nm into the solution[18]. **c**, Top, i–v schematic representation of central reaction steps during the formation of a crystalline COF framework[32]. Bottom, i–iv background-subtracted iSCAT images visualizing spatially and temporally the corresponding dynamic processes. The images show: i, pre-nucleation: the monomers in reactant solution; ii, catalyst addition: solvent restructuring and nucleation of mesitylene droplets; iii, nucleation and precipitation; iv, growth processes of solid matter during TA-TAPB COF formation (3 M HOAc, room temperature). Scale bars, 4 μm. v, exemplary PXRD pattern of a highly crystalline TA-TAPB COF (6 M HOAc, 120 °C, 72 h). To obtain highly ordered materials, the framework is typically allowed to find its optimal configuration by reformation of bonds by applying elevated temperature and pressure throughout a several-days-long solvothermal synthesis (usually 120 °C for 3 days)[13]. PXRD measurements serve, together with gas adsorption analysis, as tools to determine the quality of the formed COFs at the end of the process.

The in operando iSCAT imaging of the TA-TAPB COF formation revealed three main stages, namely, pre-nucleation, nucleation and growth (Fig. 1c,i–iv and Supplementary Video 1). To simplify the discussion on the COF assembly process, we describe these stages as distinct. However, we note that the crystallization process of COFs is complex and these stages can occur at similar timescales[23–25] (see discussion in Supplementary Information section 4).

## Liquid–liquid phase-separation processes

In Fig. 2, the temporal evolution of the overall iSCAT signal at a 2 ms to 80 s range (Fig. 2a and Supplementary Information section 3e) is correlated to its corresponding spatial evolution on the nanoscale

(Fig. 2b). Contrast changes in the background-subtracted iSCAT images correspond to changes in the local refractive index (reference background is the mixture of solute reactants at the imaging starting point).

Notably, following catalyst addition to the precursor solution (aqueous HOAc to TA-TAPB in 1,4-dioxane/mesitylene), liquid-phase-rearrangement processes at the first several hundreds of milliseconds are clearly observed, constituting the pre-nucleation stage (Fig. 2a,b,i–iv and Supplementary Video 2). iSCAT imaging resolves that, following addition, aqueous HOAc immediately accumulates at the coverglass surface, which produces a high signal at the camera (Fig. 2a,b,ii). The high signal is attributed to the increased difference in refractive index ($\Delta n$) and therefore increased reflection at the emerged glass–water interface ($\Delta n = 0.187$) compared with the glass–dioxane/mesitylene interface ($\Delta n \approx 0.091$; Supplementary Information sections 5 and 6). Subsequently, the high local polarity induced by the hydrophilic water layer results in an abrupt decrease in solubility of the hydrophobic mesitylene and its nucleation as droplets on the coverglass surface (Fig. 2a,b,iii, Extended Data Fig. 1a and Supplementary Information section 7). In the iSCAT image, the micrometre-sized droplets exhibit a dark contrast owing to the lower refractive index difference of the glass–mesitylene interface ($\Delta n = 0.017$). Thereafter, the droplets grow presumably by means of Ostwald ripening (Fig. 2a,b,iii,iv, Extended Data Fig. 1b and Supplementary Information section 8), whereas the water dissolves into the solution (Fig. 2a,b,iv). Finally, the hydrotropic 1,4-dioxane acts as a bridging solvent and interacts with both mesitylene (hydrophobe) and water (hydrophile) to homogenize the solvent mixture[26] (Extended Data Fig. 1c). The iSCAT contrast for the resulting ternary solvent system is enhanced compared with the binary, which originates from an increase in $\Delta n$ after the incorporation of water ($\Delta n \approx 0.100$; Fig. 2a,b,v and Supplementary Information section 6). Although in Fig. 2b only three images (ii–iv) are shown for better readability, they stand exemplary for restructuring processes that take place in 2.2 s (see Supplementary Figs. 7 and 23). Notably, similar solvent restructuring has been observed for other catalyst concentrations, but with different speeds (see Supplementary Information section 2b and Supplementary Videos 3 and 4).

First solid matter is detected in the form of black contrast spots directly after the phase-rearrangement processes (Fig. 2b,v), attributed to chemical reactions initiated in parallel to the solvent rearrangements. The emergence of these aggregates is initially masked for iSCAT by the high signal intensities of the solvent-restructuring processes. After the catalyst is distributed in the medium, in-solution polymerization begins, which can be traced with iSCAT by following the fluctuations in the intensity of the detected signal. These intensity oscillations are attributed to the processes of the nucleation and growth of embryonic COF seeds in the reaction medium (Extended Data Fig. 2 and Supplementary Videos 5 and 6). In the following seconds, more particles (approximately 35% iSCAT contrast; diameter less than or equal to about 100 nm; Supplementary Information section 3f) break out from the reactive solvent mixture and attach onto the coverglass surface (Fig. 2a,b,vi and Supplementary Video 6). Subsequently, the particles grow on the coverglass surface and interconnect (Fig. 2a,b,vii) to form a film (Fig. 2a,b,viii). Notably, we were able to visualize the attachment (approximately 3 s) of a single particle and its subsequent growth (approximately 50 s) by iSCAT (Extended Data Fig. 3 and Supplementary Videos 7 and 8).

## Solvent structuring in conventional COF synthesis

The type of liquid–liquid fluctuations observed during the reaction (Fig. 2b,ii–iv) mirrors those found in a homogeneous phase during surfactant-free spontaneous emulsification[26,27], implying that the ternary reaction media used in conventional COF synthesis, could—in fact—be structured on a nanoscale/mesoscale, rather than being homogenous mixtures.

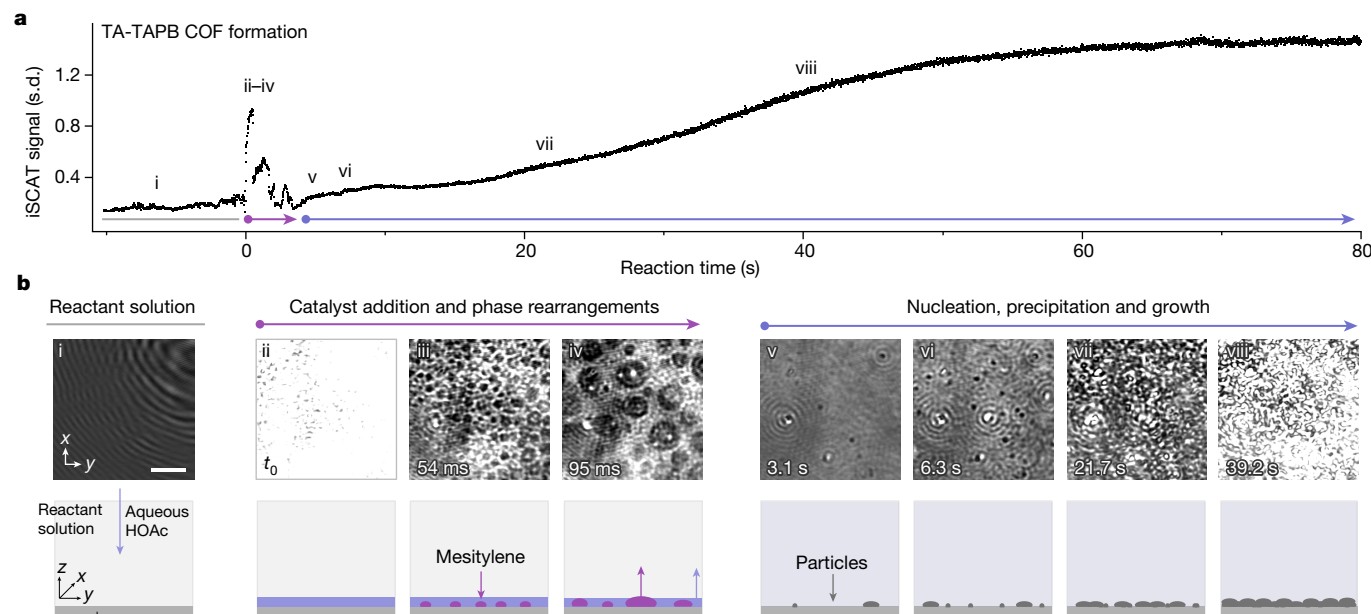

**Fig. 2 | Real-time iSCAT images visualize COF formation spatially and temporally, revealing liquid–liquid phase-separation processes in milliseconds after catalyst addition. a**, Reaction of TA-TAPB COF formation traced with iSCAT (3 M HOAc, room temperature). The graph represents the temporal evolution of the scattering in the probe region during the reaction. As such, the standard deviation (s.d.) of pixel intensities in each image is plotted as a function of time (13.5 ms per frame). The emergence and growth of solid matter results in increased scattering and increased s.d. **b**, iSCAT visualization and schematic interpretation of characteristic points in **a**. The iSCAT images provide spatial information on the reaction starting from the solute monomers (i).

The images show that catalyst addition results in nucleation, growth and subsequent dissolution of liquid mesitylene droplets (ii–iv) on a millisecond-scale. v–viii show the emergence and growth of polymer into a continuous solid film on a second timescale. During growth, the signal of the polymer turns from black to white, as the pure scattering signal starts exceeding the interferometric contribution[2]. The signal varies considerably during the reaction; therefore, the contrast has been adjusted between boxes for better visualization (normalized intensity in i and v–viii: 0–0.7 and in ii–iv: 0.36–1.00). Scale bar, 4 μm (applies to all images).

Spontaneous emulsification can occur without the aid of surfactants in solvent systems comprising two immiscible solvents (that is, hydrophile and hydrophobe) and another solvent that is sufficiently miscible with both (that is, hydrotrope)[26,28,29]. The hydrotropic solvent acts as a surfactant and is located at the interface between an oil-rich phase and a water-rich phase[30]. Apart from macroscopically visible biphasic effects (for example, the ouzo effect) and a homogenous molecular phase, there are also regions in the phase diagram of these ternary solvent systems in which the solvents are structured as nanoscale or mesoscale entities, while appearing as macroscopically homogeneous solution, termed 'pre-ouzo' or surfactant-free microemulsion (SFME)[26] (Fig. 3a). These aggregates can be oil-rich, bicontinuous (bc) or water-rich and, therefore, cultivate oil-in-water (o/w), bc or water-in-oil (w/o) emulsions[31]. The related ternary phase diagrams (shown schematically in Fig. 3a) are dependent on several parameters (see discussion in Supplementary Information section 9).

The solvent system used for the synthesis of the TA-TAPB COF fulfils the requirements for surfactant-free emulsification, with 1,4-dioxane as hydrotrope, mesitylene as hydrophobe and water/HOAc as hydrophile. Moreover, the predicted non-equilibrium processes involved in the establishment of a SFME[26,27] are visualized with iSCAT following solvent mixing (Fig. 2 and Supplementary Fig. 7). Furthermore, we observed that the phase fluctuations occur exclusively for the case of the ternary solvent system, capable of forming SFMEs, but not in the binary solvent system consisting of 1,4-dioxane and aqueous HOAc (Supplementary Information section 10 and Supplementary Video 9).

Indeed, most of the common solvent systems reported as suitable for the synthesis of (imine-connected) COFs meet the conditions of surfactant-free emulsification (see the 13 distinct solvent systems in Supplementary Table 4 and more than 30 so far unexplored solvent systems in Supplementary Table 5).

In the case of COF synthesis, a nanostructured/mesostructured solvent system can be proposed as a tool to control the initial reactant interactions, in which high reaction speed provokes kinetic defects[10,32,33] (for example, in the form of overlays, defect sites, ill-defined agglomerations, interlayer crosslinks and so on)[34]. Control over this reaction facet is expected to reduce these defects and promote the precipitation of an ordered matter. In the case of a binary solvent system, the reactive counterparts are homogenously distributed, resulting in fast kinetics and presumably increased defect concentrations (Fig. 3d; schematically in Extended Data Fig. 4a). Introducing solvent structuring with a third solvent can place a dynamic barrier between the reactants and the catalyst as they compart predominantly in the oil-rich and water-rich phases of the emulsion, respectively[35] (Fig. 3e). Notably, the reaction predominantly initiates at the interface (see Extended Data Fig. 5 and Supplementary Video 10). In the case of TA-TAPB COF formation, compartmentation is accomplished by the inclusion of mesitylene. This is accompanied by a 30-fold increase in induction period (defined as the time period between catalyst addition and integrated iSCAT signal onset; Supplementary Information section 12) and a considerably increased long-range order of the resulting COF compared with the material obtained with the binary solvent mixture (Fig. 3b). Notably, the timing of the solvent addition is critical—adding mesitylene in a later stage of the reactions afforded a poorly ordered COF material and shows that the impact of the structured solvent is mainly on the speed of the initial, rapid polymerization stage (see Supplementary Fig. 33).

The revelation of solvent structuring is the basis on which another puzzling aspect of imine COF formation can be rationalized, namely the relatively high concentration of acetic acid catalyst concentration used in most protocols (typically 6 M)[36]. Increasing the HOAc concentration from 1 M to 6 M results in a considerable increase in the initial reaction speed observed (Fig. 3c; induction period: 181 s to 16 s; Supplementary

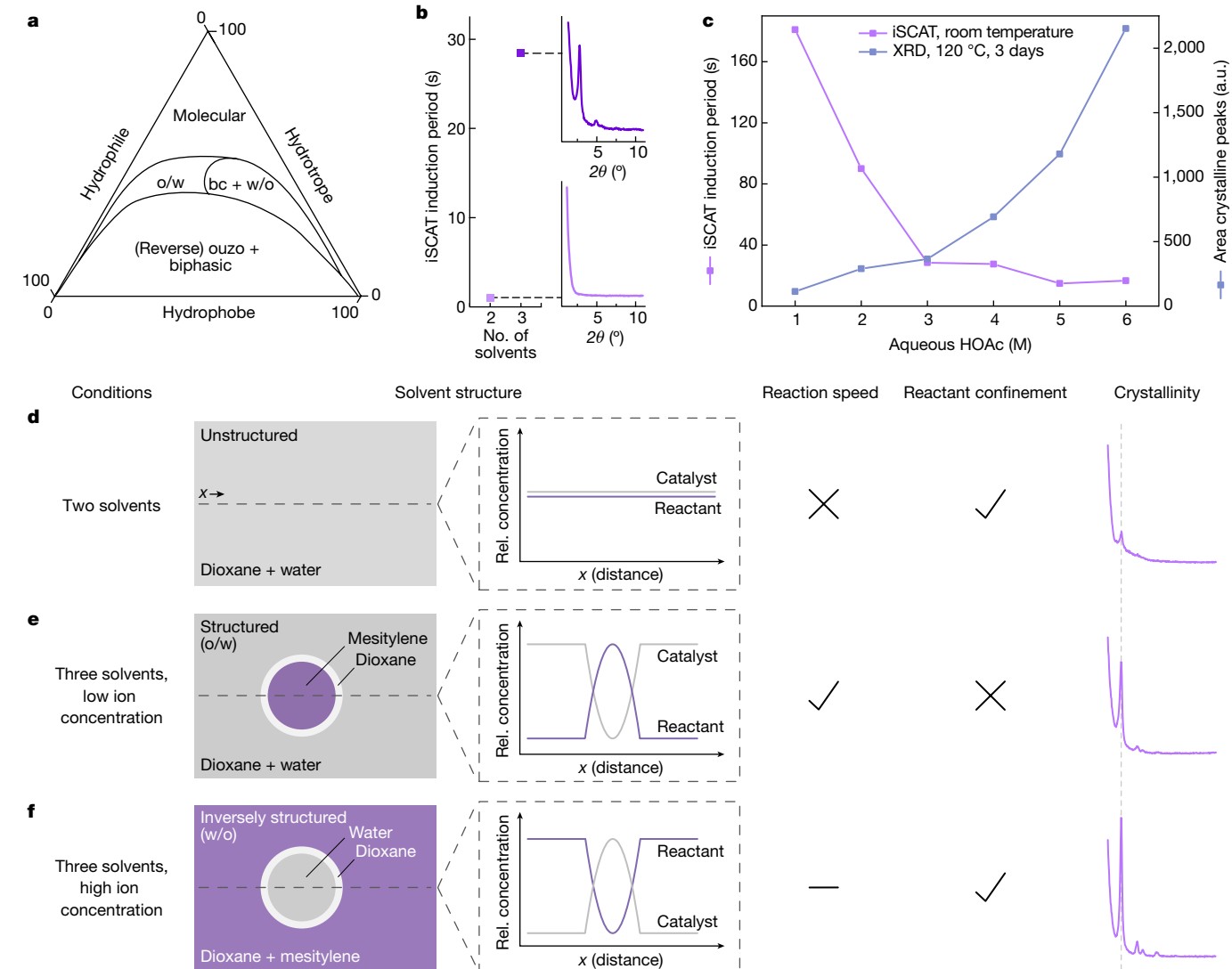

**Fig. 3 | Solvent structuring in conventional COF synthesis.** The liquid–liquid phase separations visualized in Fig. 2 originate from the combination of a hydrophobic, a hydrophilic and a hydrotropic solvent (mesitylene/water/1,4-dioxane) in the reaction. Conventional COF synthesis is executed in this type of ternary solvent system. **a**, Schematic illustration of a liquid phase diagram of these types of ternary solvent system. In the border region between molecular and biphasic solutions, macroscopically homogenous but microscopically inhomogeneous regimes exist. Here the solvents are structured as o/w, bc or w/o nanoaggregates/microaggregates. **b**, Left, induction period (measured by iSCAT) of TA-TAPB COF formation at room temperature in an unstructured binary solvent system (1,4-dioxane and 3 M aqueous HOAc) and in a structured ternary solvent system (1,4-dioxane/mesitylene/3 M aqueous HOAc). Right, the corresponding normalized PXRD patterns after 3 days show a marked enhancement in crystallinity for the COF formed in the structured solvent, attributed to slower initial polymerization and fewer kinetic defects (PXRD patterns: $2\theta = 1.2°–11.0°$). **c**, Effect of HOAc concentration on TA-TAPB COF formation with respect to solvent structuring. Higher catalyst concentrations result in a considerable decrease in the induction periods but simultaneously in the increase of crystal coherence length (qualitative visualization by means of the area under all crystalline reflections). This can be rationalized by phase-inversion processes of the structured solvent; see **f**. **d**–**f**, Schematics and normalized PXRD patterns showing the impact of solvent-structuring regimes on TA-TAPB COF formation and crystallinity (PXRD patterns: **d**,**e**, 3 M HOAc; **f**, 6 M HOAc; all, $2\theta = 1°–15°$, 120 °C, 3 days). In the case of o/w structuring (**e**), the solvent aggregates provide a kinetic barrier, in which the reactants are restricted mainly to the oil phase. Increasing the ion concentration (HOAc) induces a phase inversion towards w/o aggregates (**f**). This removes the confinement of the reactants while still providing a kinetic barrier restricting the catalyst, resulting in high crystallinity. a.u., arbitrary units.

Fig. 30), which is expected to lead to a higher amount of introduced defects. Counterintuitively, such an increase in catalyst concentration improves considerably the long-range order of the produced TA-TAPB COFs (Fig. 3c and Supplementary Fig. 34). This conundrum can be settled by ascribing the acetic acid catalyst a further role that is beneficial for the formation of an ordered framework and balances out the increased reaction speed, as explained next.

In the case of the TA-TAPB COF reaction, we propose two scenarios for the solvent structuring, promoted by the amount of solute ion concentration, here in the form of acetic acid. Low ion concentration will result in the confinement of mesitylene and thereby reactants in nanometre-sized droplet domains, as a result of the formation of o/w aggregates[26] (Fig. 3e). Such a nanoscale spatial confinement of COF reactants is thought to be counterproductive for their assembly into thermodynamically favourable states (for example, in terms of steric hindrance, concentration gradients, self-templating behaviour[10,37], diffusion and reversibility[38]). On the other hand, increasing the amount of acetic acid and, by that, the solute ions will result in a process known as salting out[39]. Here the ions assemble in the water-rich phase, in which they reduce the solubility of the hydrotrope, namely 1,4-dioxane, and

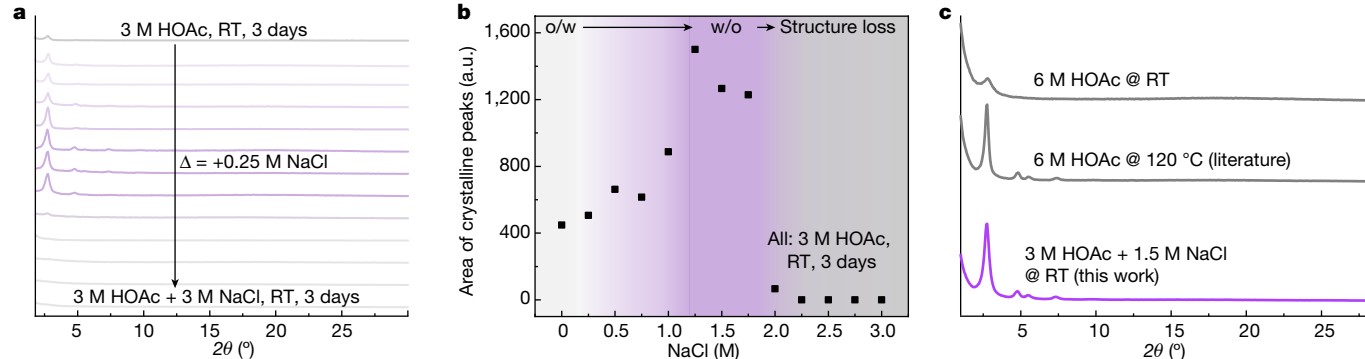

**Fig. 4 | Rationally designed room-temperature synthesis protocol for COFs. a**, Normalized PXRD patterns of TA-TAPB COFs synthesized using NaCl and acetic acid mixtures at room temperature (RT). The crystallinity of COFs obtained improves considerably with increasing NaCl concentration until a critical concentration is reached. Higher concentration of NaCl results in the collapse of long-range order of the obtained materials. The strong dependence of the material crystallinity on the ion concentration in the reaction mixture is attributed to the salting-out effect of the added ions and the corresponding change of the solvent structure. The qualitative trend of structural order is visualized in **b** by plotting the area under all crystalline PXRD reflections as a function of NaCl concentration. **c**, Normalized PXRD patterns of TA-TAPB COFs synthesized by using the IAC protocol at room temperature and by the conventional solvothermal method at room temperature and 120 °C. The TA-TAPB COF produced by IAC at room temperature exhibits similar crystallinity compared with conventional synthesis at 120 °C. In the IAC reaction scheme, acetic acid is partially replaced by sodium chloride to induce a salting-out effect while having minimal impact on the reaction speed. a.u., arbitrary units.

thereby effectively force the hydrotrope into the oil-rich phase. This results in a transition of the emulsion from o/w towards w/o aggregates, eventually leading to a phase inversion[31,40] (schematically shown in Fig. 3f). After phase inversion, the reactants are located in the continuous main phase (oil-rich), whereas the catalyst is confined in water-rich entities. Although the kinetic barrier of the emulsification is still in place (induction period 16 s versus 1 s binary/unstructured; Supplementary Fig. 35), the reactants are unconstrained and able to assemble in their thermodynamic most favourable state in the oil-rich phase, empowering the formation of an ordered framework (schematically in Extended Data Fig. 4b). This is represented as an increase in long-range order compared with a lower HOAc concentration (see powder X-ray diffraction (PXRD) patterns in Fig. 3e,f).

## Rational design of COF synthesis

To examine the suggested solvent-structuring and phase-inversion properties of the reaction mixture, we turned to a rational design of the TA-TAPB COF formation environment using low amount of catalyst compensated with salt ions. We aimed at uncoupling the two roles of acetic acid, namely the contribution to the reaction speed and its salting-out effect, by replacing a certain amount of the acetic acid with sodium chloride. Here NaCl is a tool to alter the boundaries and areas of the subregions in the phase diagram by increasing the interfacial (water–oil) surface tension and inducing a salting-out effect (mimicking HOAc)[39,40]. We termed this approach ion-assisted conversion (IAC).

After confirming the salting-out effect of NaCl with iSCAT (see Supplementary Information section 14), the COF reaction was initially carried out at room temperature using 3 M acetic acid (instead of the common 6 M), which increases the iSCAT induction time by 75% (16 s to 28 s; Fig. 3c). To the catalyst mixture, we then added complimentary Na⁺Cl⁻ ions to achieve a solute ion concentration suitable for a phase inversion. Here we confirmed that, for NaCl concentrations <2 M, the induction period is still 50% longer than for 6 M acetic acid (Supplementary Figs. 31 and 35). Evidently, the PXRD patterns obtained with increasing amount of NaCl indicate a gradual increase in the COF order for 0.25–1.25 M NaCl added (Fig. 4a,b and Supplementary Fig. 36). This increase is attributed to an emulsion phase transition from o/w crossing bc to w/o, which is associated with a gradual decrease in reactant confinement. For 1.25–1.75 M NaCl, the COF crystallinity stays at a high level. Notably, the polymerization conditions are improved in such a

way that the COF powder obtained after 3 days at room temperature exhibits similar pattern characteristics as the highly crystalline material commonly achieved after 3 days at 120 °C (Fig. 4c). Moreover, the porosity, pattern-reflection positions and overall morphology are in line with our previous report[41] of the material (Supplementary Information section 15). Notably, at ≥2 M NaCl, the crystallinity decreased abruptly, associated with weakening of all observed reflections, which we attribute to a loss in solvent structure[35]. This is supported by an iSCAT induction period of less than 1 s for 3 M NaCl added, which is slightly shorter than for the unstructured binary solvent (Supplementary Figs. 31 and 35). Ultimately, using HOAc/NaCl mixture (for example, 3 M HOAc/1.5 M NaCl) instead of 6 M HOAc, the initial speed of the TA-TAPB COF polymerization has been adjusted while a high solute ion concentration for a salting-out effect has been maintained, resulting in room-temperature COF synthesis.

Next we investigated the IAC approach in terms of required conditions, generalizability, upscaling and its effect on the COF crystallization (Supplementary Information section 16). We show that the IAC does not facilitate COF formation in a binary, unstructured solvent system and that it relies on the ability of the system to exhibit structuring. Furthermore, similar to mesitylene inclusion, the timing of salt addition (at the initial stage) is crucial for the COF synthesis, indicating that the initial reaction environment is of greatest importance. Given that these requirements are fulfilled, the IAC approach is generalizable to the concept of replacing catalyst with ions and the COF can form with different ion-to-catalyst ratios and ion types. Further, the IAC approach can be used for gram-scale synthesis, an important aspect in view of potential industrial applications. Finally, in situ X-ray diffraction (XRD) analyses indicate that, under IAC conditions, a framework comprising fewer structural faults forms in the early stages of the reaction compared with the conventional approach at room temperature.

Following the guiding principles of the IAC protocol established for TA-TAPB COF, we aimed at expanding the solvent-phase-modulation approach and implementing it in the synthesis of further COF systems. Hereby we demonstrated successful room-temperature synthesis for a total of four COF systems, two distinct solvent-mixture systems and by exploiting two types of additive (inorganic/antagonistic salts) (see Methods and Supplementary Information section 17).

In summary, we have demonstrated that direct imaging with iSCAT microscopy is a powerful tool to decode multistage wet-chemical processes such as the synthesis of COFs. The IAC protocol constitutes a

robust approach for synthesizing framework materials under mild conditions. Furthermore, the strategies outlined for customizing the COF reaction environment using liquid phase diagrams can be widely adopted in materials synthesis. Through the visualization of reaction landscapes using light-scattering techniques and the active design of solvent structures, we predict a pathway towards rational materials synthesis extending beyond framework materials.

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

## Methods

### Chemicals

All materials were purchased from Sigma-Aldrich, Acros or TCI Europe in the common purities purum, puriss or reagent grade. The materials were used as received without further purification and handled in air unless otherwise noted.

### PXRD

PXRD measurements were performed using a Bruker D8 Discover with Ni-filtered Cu Kα radiation and a LynxEye position-sensitive detector. For using PXRD patterns as a merit for the COF crystallinity, measurements were performed with identical parameters, including scan speed and intervals. PXRD patterns presented in Figs. 1c, 3b,d–f and 4b,c were measured at a scan speed of 2 s per 0.05 2θ interval. The remaining PXRD patterns presented in Figs. 3c and 4a and the Supplementary Information were measured at a scan speed of 0.5 s per 0.05 2θ interval. In case of normalization of the diffraction pattern, the point with the highest count score at the pattern has been taken as reference. In Fig. 4b, the area of all crystalline reflections was determined by analysing the respective non-normalized, background-corrected PXRD patterns.

### In situ small and wide-angle X-ray scattering

In situ measurements were performed using an Anton Paar SAXSpoint 2.0 system equipped with a Primux 100 micro Cu Kα source and a Dectris EIGER R 1M 2D detector. A reaction solution was prepared before measurement and initiated with acetic acid three minutes before the start of the in situ measurement. This time was needed for transferring the reactive solution into a quartz capillary of 2 mm in diameter and subsequently to the measurement chamber under vacuum. The quartz capillary was kept at 293.2 K during the whole experiment. The capillary was positioned at a sample–detector distance of 180 mm. The data were recorded with intervals of 10 min or 1 h, as indicated in the respective figure captions.

### Gas adsorption analysis

Nitrogen sorption isotherms were recorded with Quantachrome Autosorb-1 and Autosorb iQ instruments at 77 K. The samples were outgassed for 24 h at 120 °C under high vacuum before the measurements.

### Scanning electron microscopy

Scanning electron microscopy images were recorded with a FEI Helios NanoLab G3 UC scanning electron microscope equipped with a field emission gun operated at 3–5 kV.

### Ultraviolet–visible absorption spectroscopy

Ultraviolet–visible spectra were recorded using a PerkinElmer LAMBDA 1050 spectrometer equipped with a 150-mm integrating sphere, photomultiplier tube and InGaAs detectors. Diffuse reflectance spectra were collected with a Praying Mantis (Harrick) accessory and were referenced to barium sulfate powder as white standard. The specular reflection of the sample surface was removed from the signal using apertures that allow only light scattered at angles >20° to pass.

### Fourier-transform infrared spectroscopy

The Fourier-transform infrared spectra were recorded on a Bruker VERTEX 70 FT-IR instrument using a liquid-nitrogen-cooled MCT detector and a germanium ATR crystal. The infrared data are background-corrected and reported in frequency of adsorption (cm$^{-1}$).

### COF synthesis procedure in bulk

**TA-TAPB COF.** *Standard protocol with HOAc as catalyst (25 °C/120 °C).* The TA-TAPB COF was synthesized following a previous report[41]. In a 6-ml culture tube, TA (6.05 mg, 0.045 mmol, 1.5 equiv.) and TAPB (10.5 mg, 0.03 mmol, 1 equiv.) were suspended in 500 µl of a 9:1 (v/v) mixture of 1,4-dioxane/mesitylene. Unless otherwise stated, aqueous acetic acid in a specific concentration (6 M, 50 µl) was added, the tube was sealed and the reaction mixture was sonicated to ensure sufficient mixing of the components. Afterwards, the reaction mixture was either heated to 120 °C for 72 h or kept at room temperature for 72 h. After the given reaction time, a yellow precipitate was isolated by filtration under vacuum. Maintaining the material in a slightly wet state[24], the sample was extracted using $CO_2$ under supercritical conditions. See also our general comment about drying of the TA-TAPB COF in the following.

*Protocol with HOAc/NaCl as catalyst mixture (25 °C).* The TA-TAPB COF was synthesized following a previous report[41]. In a 6-ml culture tube, TA (6.05 mg, 0.045 mmol, 1.5 equiv.) and TAPB (10.5 mg, 0.03 mmol, 1 equiv.) were suspended in 500 µl of a 9:1 (v/v) mixture of 1,4-dioxane/mesitylene. Unless otherwise stated, aqueous acetic acid and NaCl in a specific mixture (3 M HOAc, 1.5 M NaCl, 50 µl) were added, the tube was sealed and the reaction mixture was sonicated to ensure sufficient mixing of the components. Afterwards, the reaction mixture was kept at room temperature for 72 h. Subsequently, the samples were washed with ethanol thoroughly to remove remaining NaCl. Maintaining the material in a slightly wet state[24], the sample was extracted using $CO_2$ under supercritical conditions. See also our general comment about drying of the TA-TAPB COF in the following.

*Gram-scale protocol with HOAc/NaCl as catalyst mixture (25 °C).* The TA-TAPB COF was synthesized on the gram scale according to the lab-scale procedure described above. In a 200-ml Schott Duran glass bottle, TA (1.21 g, 9 mmol, 1.5 equiv.) and TAPB (2.1 g, 6 mmol, 1 equiv.) were suspended in 100 ml of a 9:1 (v/v) mixture of 1,4-dioxane/mesitylene. Aqueous acetic acid and NaCl in a specific mixture (3 M HOAc, 1.5 M NaCl, 10 ml) were added, the glass bottle was sealed and the reaction mixture was sonicated to ensure sufficient mixing of the components. Afterwards, the reaction mixture was kept at room temperature for 72 h. Subsequently, the samples were washed with ethanol thoroughly to remove remaining NaCl and dried under reduced pressure. See also our general comment about drying of the TA-TAPB COF in the following. A yellow powder was obtained (2.49 g; 86%).

*General comment on drying of TA-TAPB COF.* According to our previous report[41], the workup procedure employed can have a big impact on the quality of the obtained TA-TAPB COF. Although we observed in this work that it is possible to obtain the same quality of material by either drying under high vacuum or sCO$_2$ extraction, we recommend sCO$_2$, as it is independent of the human factor and highly robust.

**WTA COF.** The WTA COF was synthesized following a previous report[42]. Under argon, *N,N,N′,N′*-tetrakis(4-aminophenyl)-1,4-phenylenediamine (W, 6.00 mg, 0.013 mmol, 1 equiv.) and TA (3.40 mg, 0.025 mmol, 2 equiv.) were dissolved in 1 ml of benzyl alcohol and mesitylene (9:1 or 10:10 v/v). The aqueous catalyst mixture (50 µl), including acetic acid, acetic acid/NaCl or acetic acid/Ph$_4$PCl in specific concentrations, was added and the tube was sealed. The specific concentrations are stated in the respective Supplementary Figs. After addition of the respective catalyst mixture, the reaction mixtures were kept at room temperature or at 100 °C for 3 days. The resulting red precipitates were collected by filtration, washed with tetrahydrofuran and dried under reduced pressure.

**TAPB-DMTA COF.** The TAPB-DMTA COF was synthesized according to the synthesis conditions of TA-TAPB COF. In a 6-ml culture tube, 3,6-dimethoxyterephthalaldehyde (DMTA, 4.35 mg, 0.023 mmol, 1.5 equiv.) and TAPB (5.25 mg, 0.015 mmol, 1 equiv.) were suspended in 1 ml of a 9:1 (v/v) mixture of 1,4-dioxane/mesitylene. The aqueous catalyst mixture (50 µl), including either acetic acid or acetic acid/NaCl in specific concentrations, was added and the tube was sealed. The specific concentrations are stated in the respective Supplementary Figs. Afterwards, the reaction mixture was kept at room temperature or 120 °C for 72 h. Subsequently, the samples were washed with ethanol

thoroughly to remove remaining NaCl, then with tetrahydrofuran and subsequently dried under reduced pressure. The TAPB-DMTA COF was obtained as yellow powder.

**TT-ETTA COF.** The TT-ETTA COF was synthesized following a previous report[43]. Under argon, 1,1,2,2-tetra(4-aminophenyl)ethene (ETTA, 5.85 mg, 0.015 mmol, 1 equiv.) and thieno-[3,2-*b*]thiophene-2,5-dicarboxaldehyde (TT, 5.9 mg, 0.030 mmol, 2 equiv.) were dissolved in 500 µl of a 9:1 (v/v) mixture of benzyl alcohol and mesitylene. To test the impact of NaCl on the formation of this COF, three different catalyst mixtures were prepared: (1) aqueous acetic acid (6 M, 50 µl); (2) aqueous acetic acid (3 M, 50 µl); and (3) aqueous acetic acid and salt solution (3 M HOAc, 1.5 M NaCl, 50 µl). After addition of the respective catalyst mixture, the reaction mixtures were kept at room temperature for 3 days. The resulting red precipitates were collected by filtration, washed with acetonitrile and dried under reduced pressure. Further, the samples were extracted using $CO_2$ under supercritical conditions to remove solvent residuals.

## Operando iSCAT measurements
The synthesis protocol of TA-TAPB COFs on the iSCAT microscope followed the bulk protocols described above and the reaction mixtures were prepared in the same way and at the same concentrations. Unless stated otherwise, for the measurement on the microscope, 100 µl reactant solution (TA-TAPB in 1,4-dioxane/mesitylene) and 10 µl aqueous catalyst mixture (for example, HOAc or HOAc/NaCl) were used. A custom-made reaction cell made from Teflon served as a reactor. In a typical experiment, first the reactant solution was added to the cell. Next, image acquisition is started. After acquiring a sufficient number of images as reference for subsequent background subtraction, the catalyst mixture is added to initiate the reaction.

## iSCAT contrast for determining the size of the scatterer
The spatial resolution of an optical technique such as iSCAT is diffraction-limited[44]. Therefore, in iSCAT, the signal of a subwavelength particle is detected on the camera as a point spread function (PSF). The PSF closely resembles a 2D Gaussian function with a full width at half maximum (FWHM) of roughly half the wavelength of the incident light (that is, $FWHM_{PSF} \gg d_{scatterer}$)[2]. However, in iSCAT, the size of the particle can be retrieved from the contrast (amplitude) of the PSF, which scales with the polarizability and therefore with the volume of the scatterer[15]. As such, despite being an optical microscope with diffraction-limited resolution, for example, the signal of a 2-nm gold particle can be detected as a PSF in the camera and by analysing its contrast, the size can be determined[45]. Typically, the contrast–size relationship is calibrated by measuring samples of known sizes[15] or through theoretical simulations/calculations[46,47].

## iSCAT as a method for imaging chemical reactions
Notably, we show that the TA-TAPB COF sample system is not interacting with the illumination light through optical absorption during iSCAT monitoring, accounting for a truly non-invasive measurement (Supplementary Fig. 14). Another important aspect for imaging reactions and the resulting products with iSCAT is that, after a certain film thickness is reached, the camera saturates owing to the high reflection signal. Although this is not relevant for the initial reaction stages, to assess the final growth state, we adjusted the absolute volume of reactant solution (100 µl) to prevent camera saturation. Finally, it has to be noted that the signal contrast in iSCAT relies on refractive index changes rather than chemical specificity, necessitating careful consideration and controls, as reflected throughout the discussion.

## Optical setup
The home-built iSCAT microscope was constructed following the instructions in refs. 48,49. The beam path and the different optics used in the setup are described in the following (see also Supplementary Fig. 1). A 785-nm single-mode laser (TOPTICA, iBeam smart 785) acts as the illumination source. The laser is coupled into a single-mode fibre (Thorlabs, custom-made for high-power applications) and is then collimated by a 10× objective (Olympus, Plan N, NA = 0.25). An achromatic plano-convex lens (Thorlabs, AC508-400-AB, $f$ = 40 cm), the focusing lens, focuses the beam into the back-focal plane of a 60× oil-immersion objective (Olympus, PLAPON60X, NA = 1.42). After passing the focusing lens, the beam is directed into the vertically positioned objective by a 45° mirror. The laser beam is collimated by the objective and illuminates the substrate. The backscattered and backreflected light of the illumination beam are then again collected by the oil-immersion objective. After passing the 45° mirror, the reflected and scattered light are sent into the detection arm by a 50/50 beam splitter. In the detection arm, a plano-convex lens (Thorlabs, AC508-1000-A, $f$ = 100 cm), the imaging lens, is installed $f$ = 100 cm after the back-focal plane of the objective and $f$ = 100 cm before a CMOS camera (Photonfocus, MV1-D1024E-160-CL). Therefore, the backscattered light that was collimated after passing the oil-immersion objective is getting focused on the camera detector by the imaging lens. By contrast, the backreflected light that was in focus at the back-focal plane of the oil-immersion objective and was defocusing thereafter gets collimated by the imaging lens and arrives as such at the camera. The signal is emerging as modulation of the reflected light by constructive or destructive interference of the scattered light with the reflected light on the camera. Acquisition software with live processing was custom-made in LabVIEW NXG.

By using a 60× Olympus objective (60× is defined relative to a standard tube lens of $f$ = 18 cm for Olympus objectives) with a $f$ = 100 cm imaging lens, the setup magnification amounts to 333 $\left(60 \times \frac{100 \text{ cm}}{18 \text{ cm}} = 333\right)$. One pixel on the detector of the Photonfocus camera has a size of 10.6 µm × 10.6 µm. Considering the magnification of 333, one pixel in a measured image (= one camera detector pixel) corresponds to 31.8 nm $\left(\frac{10.6 \text{ µm}}{333} = 31.8 \text{ nm}\right)$ on the sample/coverglass surface. The standard field of view for images taken this work was 512 × 512 pixels or 16.3 µm × 16.3 µm on the coverglass.

## Image acquisition and analysis
In this study, we used the open-source image software Fiji, based on ImageJ, for image-processing purposes. All iSCAT images presented were background-corrected by adopting the temporal median approach. To accomplish this, we calculated the median pixel intensity for each pixel across the first 300 raw images, which were captured at successive time points before the catalyst was added. Unless stated otherwise, images were acquired at a size of 512 × 512 pixels and spatially binned (2 × 2 pixels, resulting in an effective pixel size of 63.6 nm per pixel). Images presented in Figs. 1c and 2b were acquired at a speed of 2.7 ms per frame (exposure time of 1 ms, 370 fps) and five successive frames were temporally averaged, resulting in an effective temporal resolution of 13.5 ms per frame. Experiments presented in Extended Data Fig. 1 were acquired at a speed of 4.7 ms per frame (exposure time of 3 ms, 212 fps). Images presented in Extended Data Figs. 2 and 3 were acquired at a size of 400 × 400 pixels and at a speed of 2.05 ms per frame (exposure time of 1 ms, 488 fps). For images presented in Extended Data Fig. 3c, 100 frames were temporally averaged, resulting in an effective temporal resolution of 205 ms per frame. Images presented in Extended Data Fig. 5b were acquired at a size of 400 × 400 pixels and at a speed of 6.05 ms per frame (exposure time of 5 ms, 165 fps).

## Data availability
The data underlying all figures in the main text and the Extended Data are publicly available at https://doi.org/10.5281/zenodo.10947997 (ref. 50).

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

**Acknowledgements** C.G.G., L.F., R.G., D.D.M. and E.C. acknowledge financial support for seeding collaborative projects from the investment fund of the Center for NanoScience (CeNS). L.F., R.G. and D.D.M. acknowledge financial support from the Free State of Bavaria through the research network 'Solar Technologies Go Hybrid', the Deutsche Forschungsgemeinschaft (DFG, German Research Foundation) in the COORNETs SPP 1928 project ME 4515/1-2. C.G.G. and E.C. acknowledge financial support from the DFG under Germany's Excellence Strategy, EXC 2089/1-390776260, e-conversion Excellence Research Cluster, the Bavarian programme Solar Energies Go Hybrid (SolTech) and the European Commission through the ERC Starting Grant CATALIGHT (802989). We thank A. Mancini for helpful discussions during the initial phase of the project and F. Gröbmeyer for assistance with theoretical simulations. We thank A. Mancini, F. Gröbmeyer, A. Gemeinhardt, V. Sandogdhar, H. Gruber and the LMU mechanical workshop for helpful discussions about the optical setup. We thank T. Bein for his support, M. Döblinger for helpful discussions and M. Schönherr and M. Wiedmaier for technical assistance in the wet chemical laboratory.

**Author contributions** C.G.G., D.D.M. and E.C. conceived and structured the project based on the idea of C.G.G. E.C. and D.D.M. supervised and financed the project. C.G.G. designed the optical setup and planned, carried out and analysed the optical measurements. L.F. and R.G. carried out the bulk synthesis and analysis measurements and R.G. the in situ small-angle X-ray scattering measurements. All authors discussed the results. C.G.G., D.D.M. and E.C. wrote the manuscript.

**Competing interests** The authors declare no competing interests.

**Additional information**
**Correspondence and requests for materials** should be addressed to Dana D. Medina or Emiliano Cortés.

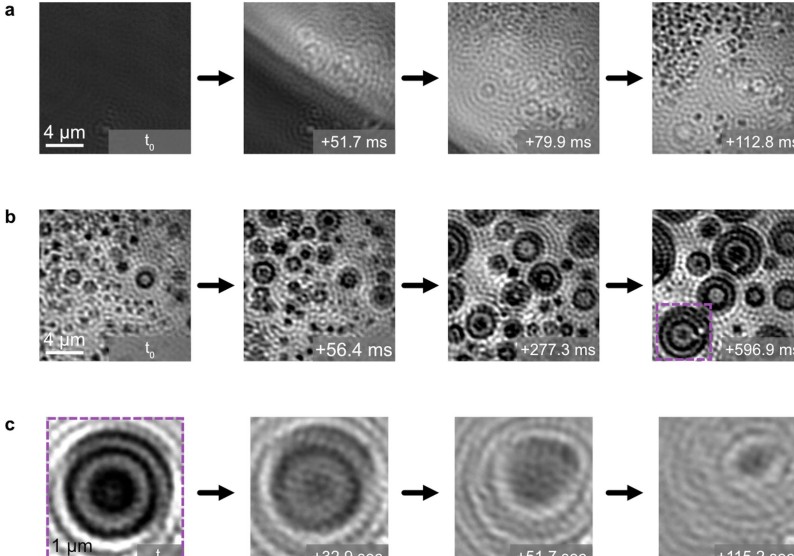

**Extended Data Fig. 1 | Nucleation, Ostwald ripening and dissolution of mesitylene droplets.** Here water (1 equiv.) was added to a model solution comprising 1,4-dioxane/mesitylene (v/v 8:2; 100 μl) as solvent system and TA (1.21 mg, 0.009 mmol) as solute. **a**, After addition of water, a white front with high iSCAT signal propagates from the upper-right corner of the image, indicating the distribution of water on the coverglass surface. Concurrently, black spots and out-of-focus PSFs emerge within the white contrast medium and attach to the coverglass surface. This phenomenon corresponds to the nucleation of hydrophobic mesitylene, facilitated by the high-polarity environment induced by the presence of water. **b**, The dark contrast mesitylene droplets grow presumably by means of Ostwald ripening. **c**, After their growth stops, the droplets reach a stable state for a limited time period. Subsequently, solvent mixing proceeds, in which 1,4-dioxane aids in the dissolution of water throughout the entire volume. Consequently, the polarity of the environment decreases, leading to a gradual dissolution of the mesitylene droplets. These interdependent processes manifest in the diminishing white signal surrounding the dark droplet (corresponding to water dissolution) and the reduction in size and decrease in black contrast of the droplet itself until complete dissolution (corresponding to mesitylene dissolution). The resulting ternary solvent system shows an enhanced contrast compared with the initial binary one owing to inclusion of water.

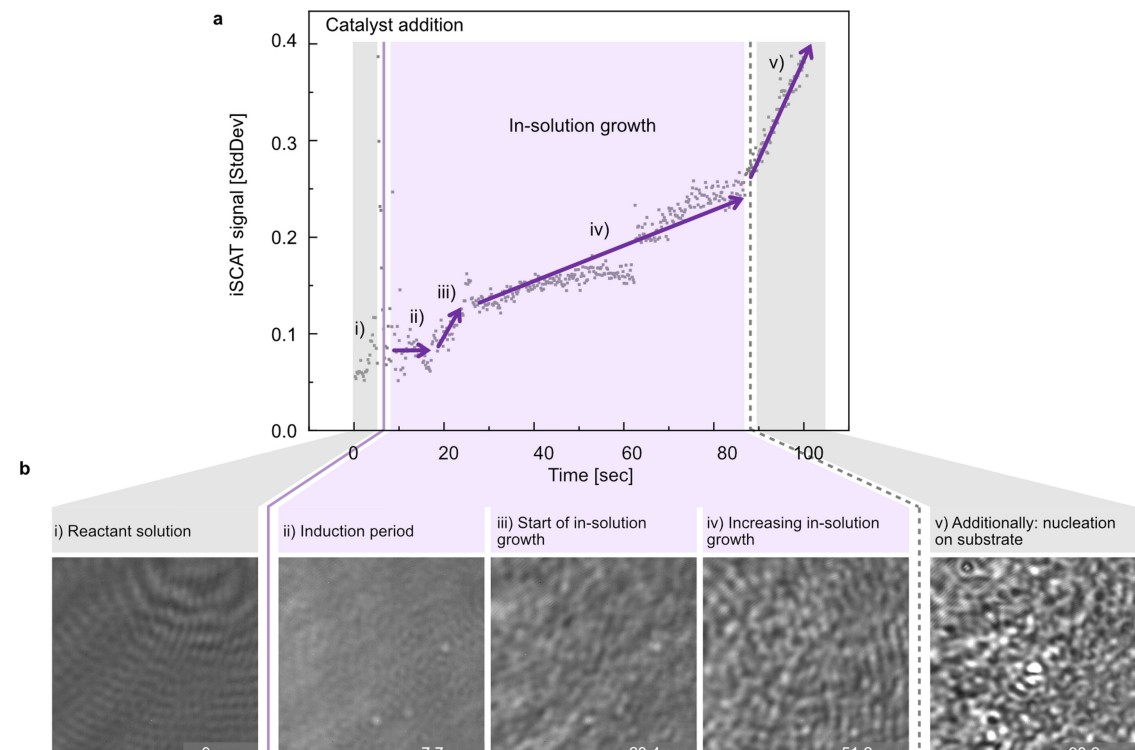

**Extended Data Fig. 2 | In-solution growth of TA-TAPB COF traced by iSCAT.** Temporal (**a**) and spatial (**b**) evolution of the iSCAT signal during the formation of the TA-TAPB COF (4 M HOAc, room temperature). Following catalyst addition to the reactant solution (i), the in-solution growth can be tracked through fluctuations in the iSCAT signal intensity. Following an induction period (ii), in-solution nucleation and growth begins (iii), resulting in a rapid increase in the iSCAT signal owing to fluctuations within the solution. These fluctuations are attributed to the presence of chemical transformations and the emergence of oligomers. The growth of solute oligomers continues, leading to a constant increase in the iSCAT signal (iv). Subsequently, solid breaks out from solution onto the surface, resulting in a sharp increase in the iSCAT signal (v). The contrast is adjusted to 0.26–2.29. Scale bar (applies to all images), 4 μm.

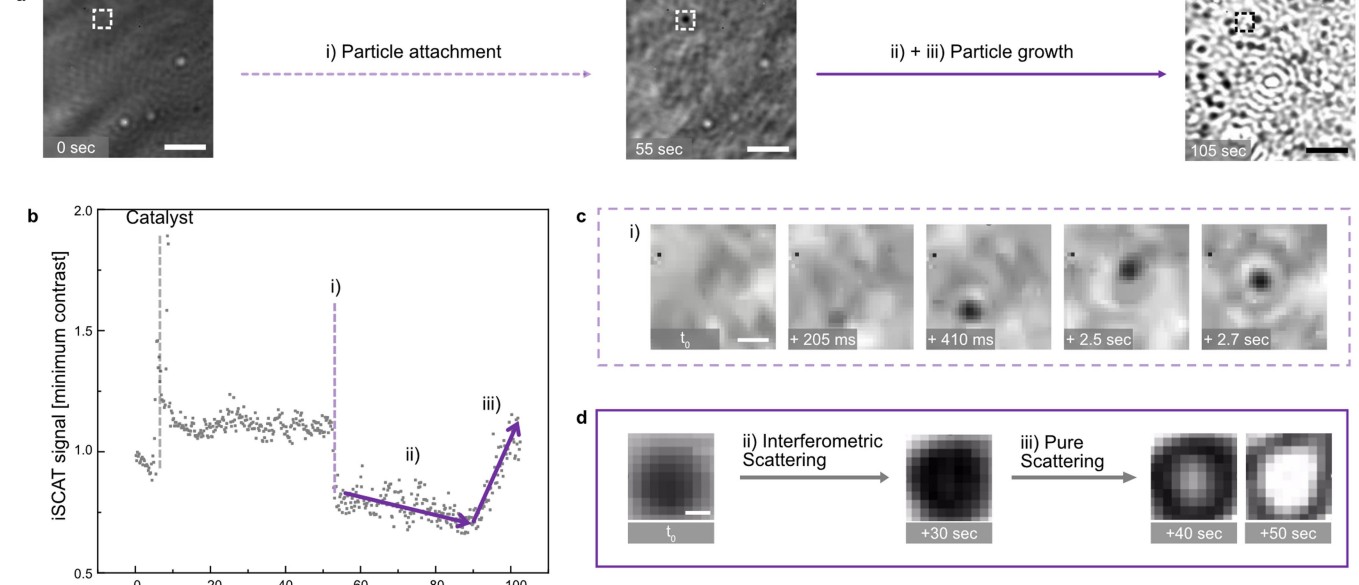

**Extended Data Fig. 3 | Attachment and growth of single nanoscale particles.**
**a**, iSCAT images captured during the formation of the TA-TAPB COF (4 M HOAc, room temperature). The white rectangle indicates the region in which the attachment (i) and growth (ii and iii) of a single particle is spatially and temporally tracked over a duration of 105 s. Scale bar (applies to all images), 4 μm. **b**, Temporal evolution of the minimum iSCAT signal within the white rectangle of **a**, starting from the solute reactants. A substantial signal increase illustrates the phase rearrangements following catalyst addition, followed by an induction period, before a sharp signal drop occurs because of the attachment of a nanoscale particle on the substrate (i) and its corresponding negative (black) contrast

modulation. Subsequently, a further signal decrease is observed owing to the growth of the seed and the increase in negative interferometric signal (ii). After reaching a certain size threshold, pure scattering becomes dominant, leading to a positive (white) contribution to the signal, resulting in an increase in contrast (iii). **c,d**, The corresponding spatially resolved maps. Diffusion and attachment of the particle from solution within a 3-s timeframe are shown in **c** (scale bar (applies to all images), 300 nm; contrast has been adjusted to 0.79–1.5). **d**, Subsequent growth of the seed and the corresponding contrast changes (scale bar (applies to all images), 200 nm; contrast has been adjusted to 0.68–1.67).

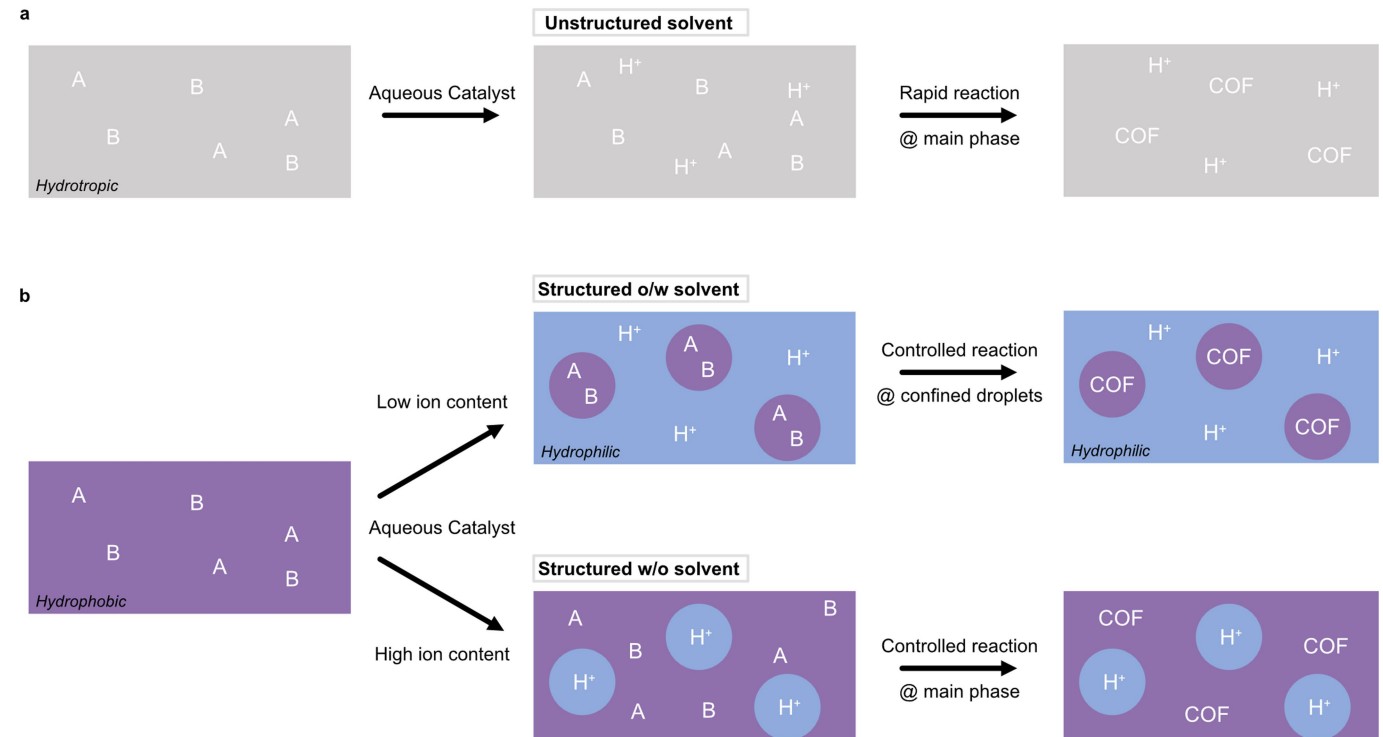

**Extended Data Fig. 4 | Schematic illustration of TA-TAPB COF formation landscapes in different solvent-structuring regions. a**, Addition of the aqueous catalyst (that is, HOAc) to a hydrotropic solvent (that is, 1,4-dioxane) including the reactants A and B (that is, TA and TAPB) yields a binary, unstructured solvent. The reaction progresses rapidly in the whole volume, inducing kinetic defects. **b**, Addition of the aqueous catalyst (that is, HOAc) to a hydrophobic solvent mixture (that is, mesitylene/1,4-dioxane) yields a ternary, structured solvent that kinetically controls the reaction. The ion concentration is a tool to govern the development of o/w or w/o solvent aggregates in the surfactant-free emulsion. For o/w emulsions, reactants are confined within aggregates; for w/o emulsions, reactants reside in the main phase, resulting in increased freedom in the assembly process and, consequently, enhanced crystallinity.

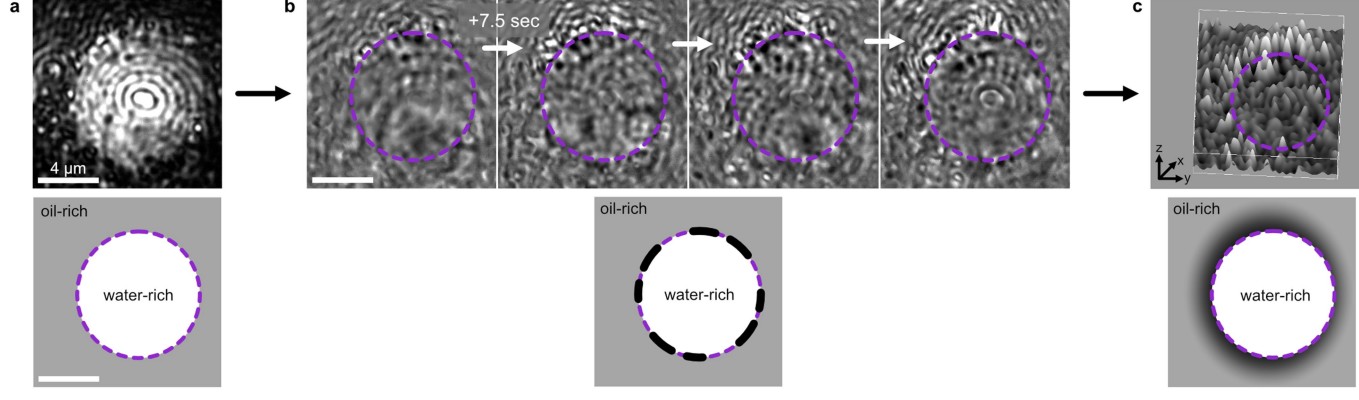

**Extended Data Fig. 5 | iSCAT imaging of a TA-TAPB COF reaction at the oil–water interface in a model system.** To examine the compartmentation of the reactants in different solvent regimes, we created a model solvent system (fivefold higher concentration of aqueous solution), exhibiting stable water-rich droplets on the coverglass surface surrounded by a continuous oil-rich phase. Here aqueous 0.06 M HOAc (5 equiv.) was added to 1,4-dioxane/mesitylene (v/v 9:1; 200 µl) including the reactants TA (2.42 mg, 0.018 mmol) and TAPB (4.2 mg, 0.012 mmol). **a**, Following catalyst addition, stable, water-rich droplets accumulated on the surface (white contrast region). The droplets are surrounded by a continuous oil-rich phase (black contrast region). The low catalyst concentration resulted in a slow reaction and long induction period (hours). The iSCAT image in **a** is taken at a point in time at which already initial reactions have taken place, visible in the high-contrast white spots in the oil-rich phase from nucleated solid. **b**, For these iSCAT images, image **a** is subtracted as background (versus using the initial state before aqueous catalyst addition as background) to show the progression of the iSCAT signal from this point in time. Increasing iSCAT signal fluctuations are detected at the interface of oil-rich and water-rich phases resulting from the emergence and nucleation of solid phase from a reaction at the interface (see also Supplementary Video 10). The contrast has been adjusted to 0.38–1.91. Scale bar (applies to all images), 4 µm. **c**, 3D surface plot of the iSCAT image after 30 s (using **a** as reference and as background image). The signal originating from nucleated solid is mainly located in the oil-rich phase, pointing to the compartmentation of catalyst (water-rich) and reactants (oil-rich).