## [Peer Review File · Nature]

Manuscript Title: Early stages of covalent organic framework formation imaged in operando

Reviewer Comments & Author Rebuttals

Reviewer Reports on the Initial Version:

Referees' comments:

Referee #2 (Remarks to the Author):

Reviewer Report: Operando Reaction Imaging Demystifies Covalent organic Framework Formation

A. Summary of the Key Results

In the submitted manuscript entitled “Operando Reaction Imaging Demystifies Covalent organic Framework Formation”, by Medina, Cortes, and coworkers, the authors use interferometric scattering microscopy (iSCAT) to explore covalent organic framework (COF) formation dynamics. The authors claim that structured solvents (i.e. emulsions) are a prerequisite for the formation of well-defined COF crystallites. The authors show this by comparing iSCAT measurements with ex situ characterization.

B. Originality and Significance

This work represents using a new technique (iSCAT) to tackle the scientific area of COF formation dynamics. I am especially fond of this work because it uses a technique that accesses the ~10s of nanometer to micron scale, which other in situ techniques often do not capture. However, this new work ignores other 2D COF formation mechanistic insight provided by others (Dichtel, Banerjee, Zhao), which I believe to be equally important in this sphere. As brief examples, Zhao and coworkers have recently obtained large crystallites using supercritical CO₂ as a solvent (Nat. Commun. 2022). This work certainly does not have phase separation occurring, which the current report discloses as a requirement. Additionally, the amorphous-to-crystalline transformation that the authors rely on has recently been challenged by its original proposers (Dichtel and coworkers), which discloses a significantly more complex pathway than what was previously assumed (J. Am. Chem. Soc. 2022). These efforts also do not fully appreciate the computational insight provided to general COF syntheses (Bredas and coworkers, J.

Am. Chem. Soc. 2020, 142, 3, 1367–1374, ACS Materials Lett. 2021, 3, 4, 398–405). Moreover, it is not clear to me how these findings align with formation dynamic studies of metal-organic frameworks, which have also been studied in-depth.

As a final note, I mention that the multi-phase nature of COF syntheses was noted in Yaghi's original 2005 report to be important for obtaining crystalline materials. They attribute this to the slowed solubility and kinetics of monomer addition, which seems to be a similar argument to the one made here.

I believe that to understand the insights provided by this report, they must be better contextualized with what has already been done in the field. Other mechanistic reports should also be discussed.

Adding salt to modulate the phases of the liquid solution is interesting and novel.

C. Data & methodology: validity of approach, quality of data, quality of presentation

The mechanistic insight provided by iSCAT is interesting. However, the inability to probe the nanostructure of the COFs limits some of the insight provided here. I know the authors collect ex situ PXRD – but I wonder if it would be possible to do in situ PXRD in a capillary condition. I am familiar with the challenges of these measurements, but I think they would greatly augment the findings presented here. It would not be critical to do this on every set of conditions, just the optimized ones (and perhaps unoptimized for comparison). This would remove some of the ambiguities that can be introduced during a work-up step.

I think that the analysis likely leads the authors to the correct conclusion, but formally, the area of the peak is not a valid method to determine crystallinity. Scherrer broadening should be considered a more accurate measure of crystallinity. This compensates for the different overall amount of material (and differences in the X-ray optics) that might vary between measured samples.

Overall, the presentation of the data is high-quality.

D. Appropriate use of statistics and treatment of uncertainties

I believe the uncertainties are well-treated.

E. Conclusions: robustness, validity, reliability

I find the use of iSCAT interesting and potentially useful. However, I am not sure that the conclusions of this paper are supported by the data provided.

The authors claim that the nanostructured solvent is required for effective COF syntheses. There are examples where this is not true. Moreover, the authors do not obtain large singlecrystals with these conditions, which other groups are now achieving with regularity.

The authors claim that these conditions can be used for any COF. They have not shown that this is true beyond a small number of prototypical systems. Perhaps the authors can expand the number of achievable systems.

F. References: appropriate credit to previous work?

Partially. See above.

Referee #3 (Remarks to the Author):

It is so rare to have the opportunity to review a manuscript in which the authors demonstrate a true advance that blends fundamental measurements and derived physical chemistry insight with actionable design principles to optimize synthetic procedures, but this manuscript does exactly that. In “Operando reaction imaging demystifies covalent organic framework formation” Gruber and colleagues employ iSCAT imaging that is sensitive to refractive index differences in a solution to image the TA-TAPB COF

formation reaction steps of pre-nucleation, nucleation and growth, critically revealing the formation and evolution of emulsion compartments that slow the kinetics of the reaction by limiting the catalyst availability. It's the sort of result where in retrospect we should have realized that there should be a connection to surfactant-free emulsions, but the authors are providing a big service to the community for not only planting this idea but also for generalizing it in a profound way: the imaging tells a lot already, but they proceed to show a series of 'salting-out' control experiments that leverage the ternary solvent diagram in question that ultimately lead to not only an optimization of the amount of acetic acid catalyst needed but to demonstrated predictive power to generalize the principle with another COF and to completely eliminate the need for 3 days of thermal annealing, allowing the annealing to form highly crystalline COFs strictly at room temperature! It is also very compelling to see that the novel iSCAT approach is properly correlated to the outcomes of standard post-anneal characterization methods. This really firmly introduces iSCAT as a tool to advance materials research. In my view, this manuscript is a landmark work that is highly worthy of publication in Nature. They have really taken the scientific method full circle, which is refreshing, and this has been a resoundingly successful multidisciplinary collaboration. Furthermore, the work is very carefully done and presented, with very strong experimental backing to the scientific rationale via a comprehensive set of control experiments.

More minor comments:

1. It would help to elaborate in the main text about how the authors know the sizes of the objects they refer to seeing in their movies e.g. 50 nm is mentioned, but the imaging is diffraction limited (might be wise to cite original iSCAT studies explaining contrast changes as a function of size on p6 line 116?), and also how they know they're seeing something floating in the solution vs on the surface of the coverslip. Furthermore, what are the requirements/constraints on imaging other reactions in the same way.
2. Fig 1c stages are not numbered but referred to with Roman numerals in the caption and text
3. Fig 2b: consider showing up to 2.2 s rather than only to 95 ms to clarify what happens between these two times
4. The Presentation of Fig 3b with 'left' and 'right' is a bit awkward. It's hard to understand what is left vs right. Perhaps box left and right separately?
5. Recheck ref 45.

6. I really appreciate the many montages of the movies that are included in the supplement. I wonder if some arrowheads could be added to point to specific key events within these montages to help the reader follow how the movies are interpreted.

7. Similarly, the extended data figures are extremely helpful to follow the narrative. Consider referring to them/describing them more explicitly rather than only parenthetically, since this will encourage the reader to consult them and enhance the paper's readability.

Author Rebuttals to Initial Comments:

Point-by-Point Response to Reviewers

We would like to acknowledge the fair assessment and interesting comments provided by the Reviewers while evaluating our manuscript. The comments received undoubtedly assisted us to advance our original manuscript further. Throughout this letter, we will present **in blue** our responses to every point raised by the Reviewers, while the modifications to the original manuscript and supplementary information are shown **in red**.

Referee #2 (Remarks to the Author):

Reviewer Report: Operando Reaction Imaging Demystifies Covalent organic Framework Formation

We would like to summarize - before discussing the comments of Reviewer #2 in detail - that we have now: a) contextualized our results within all the body of literature suggested by the Reviewer; b) performed in-situ XRD measurements to shed light on the role of the salt on the crystallization process by a complementary technique; c) demonstrated our solvent phase modulation approach for a total of four COFs systems, two solvent mixture systems and two types of additives; and d) performed additional iSCAT measurements visualizing the implications of solvent structuring.

A. Summary of the Key Results

A.1 In the submitted manuscript entitled "Operando Reaction Imaging Demystifies Covalent organic Framework Formation", by Medina, Cortes, and coworkers, the authors use interferometric scattering microscopy (iSCAT) to explore covalent organic framework (COF) formation dynamics. The authors claim that structured solvents (i.e. emulsions) are a prerequisite for the formation of well-defined COF crystallites. The authors show this by comparing iSCAT measurements with ex situ characterization.

While this description is almost perfect, there is one point we would like to specify: We detected the typically employed reaction media for the conventional solvothermal synthesis to be structured. Essentially, the default reaction procedure makes use of structured solvents as preliminary kinetic control to enable the synthesis of crystalline COFs, a factor that thus far went undetected. Indeed, despite the multiple ways of synthesizing COFs – including milling, ionothermal or supercritical CO₂, among others - in the time frame of almost 20 years, the solvothermal approach turned out to be the most robust and most used one. In order to adequately deliver our message regarding this point, we replaced in the text the word "prerequisite" with a more suitable description, as shown below:

Abstract:

Our study visualizes liquid-liquid phase separation, pointing to the existence of structured solvents in the form of surfactant-free (micro)emulsions in conventional COF synthesis. Revealing this fundamental factor illuminates the pathway towards rational COF synthesis, which we demonstrate by systematically developing a protocol at room temperature instead of elevated temperature. iSCAT microscopy offers eyes for understanding wet-chemical reactions in material science.

Introduction:

This model system is representative of the most studied COF family (>200 imine-bond connected COFs)^{6,13} and of the conventional solvothermal COF synthesis approach in ternary solvent systems.³³ Moreover, the reaction is carried out in the most-employed solvent system (1,4-dioxane/mesitylene/aqueous acetic acid).³⁴

Conclusion:

The versatility of this cost-effective and easy-to-use device^{70,71} allowed for uncovering an unknown stage in the conventional (imine) COF formation mechanism – solvent structuring.

B. Originality and Significance

We discuss each point of this section one by one, but we want to highlight that all the changes are reflected together in the manuscript and supplementary information, as shown in detail in B.2.

B.1 This work represents using a new technique (iSCAT) to tackle the scientific area of COF formation dynamics. I am especially fond of this work because it uses a technique that accesses the ~10s of nanometer to micron scale, which other in situ techniques often do not capture.

We thank the Reviewer for pointing to the novelty of using iSCAT for the investigation of complex wet-chemical reactions such as COF formation. We additionally want to emphasize that – besides accessing the 10s nm to μm -scale, we target a temporal blindspot of the first milliseconds to seconds, particularly during the phase of solvent mixing and initial nucleation and growth, which has not been accessible so far. The ability of iSCAT to combine high temporal sensitivity with the ability of tracking all matter present in the reaction by a contrast giving mechanism, is indeed unique compared to other in situ techniques and one of the top highlights of our work here, exemplified with the synthesis of COFs.

B.2 However, this new work ignores other 2D COF formation mechanistic insight provided by others (Dichtel, Banerjee, Zhao), which I believe to be equally important in this sphere.

We thank the Reviewer for this important point. We show below the new references included and the associated discussion about state-of-the-art knowledge on COF formation; but before that, please allow us to go into a bit more detail about this point.

We see our findings as complementing the existing crystallization picture by elucidating a crucial missing piece in the landscape of COF formation. In addition to the introduction of a new technique to trace the COF formation processes in operando, our approach distinguishes itself from the previous work by shedding light on the reaction stages happening before the typical mechanistic studies start: with the solvent mixing. This holistic approach enabled by the use of iSCAT, provided us the insight to elucidate the role of solvents in the COF formation stages.

However, of course, the very first stages of solvent mixing and structuring can have an impact on the successive stages of COF formation. In view of this we agree that the acquired knowledge about COF crystallization mechanism should be discussed in more depth, which indeed helped us now to contextualize our findings.

All comments of this section (**B.1 to B.9**) are summarized in the changes to manuscript and SI shown below and will be elaborated in greater detail in the following.

We added a whole, new section (Supplementary Information section 1) to the SI:

Section 1. COF crystallization mechanism in current literature

The way COFs materialize as crystalline and porous materials, is a complex process to trace. The conventional techniques to elucidate the processes, especially at the early stages of molecular interactions and in operando, are lacking features such as time resolution necessary for obtaining knowledge, especially on the time scales below 1 min.¹ Due to this, there are still open questions and vibrant discussions regarding the way the crystallization proceeds exactly in the conventional solvothermal autoclave reaction. Furthermore, theoretical predictions try to address this complex point for experimentalists by providing a computational reasoning to COF synthesis.^{2,3}

Initially, it was almost a consensus for (imine) COFs that during the fast-initial polymerization stage (condensation reactions of the monomers) an amorphous polymer is produced first which subsequently, develops long-range order through defect healing by dynamic covalent chemistry (DCC) over a longer time span of days.^{4,5} However, new evidence has been reported recently by Dichtel, Evans, Medina et al. that paint a somewhat more complex picture of the COF product crystallinity - while not discarding the overall mechanism. Medina et al. and then several other groups showed that regardless of the way crystallization proceeds during the reaction time the product isolation procedures such as vacuum activation and solvent washing can inflict harm on the long-range order of the COFs obtained, obscuring parts of the previously drawn conclusions.⁶⁻⁸ Following this, it has been reported by Dichtel et al., after exploiting gentler activation techniques, that the very initially precipitated species are, at least partially formed as a few-layer disorganized crystalline sheets. In situ synchrotron XRD measurements confirmed that crystalline matter can emerge much more rapidly than previously anticipated and that at the earliest possible measurement time of 90 sec crystalline material exists in the mixture (TA-TAPB COF).⁸

In a later paper,⁹ Zhao et al. describe the crystallization process by nucleation and growth stages. At first, a crystalline COF phase can rapidly develop from self-templated monomers along with an amorphous phase which is transferred later by a self-healing growth stage to a crystalline phase. Thereby, the overall crystallinity increases mainly owing to the conversion of the amorphous phase to a crystalline phase. This slower amorphous to crystalline transformation has been reported on several occasions.^{1,8-10} It is considered that the process is facilitated by the DCC of the initial defective material into longer-range ordered COFs (which are more robust towards the conventional vacuum activation).

In a Review by Evans, et al. it has been proposed that „a combination of mechanisms is active in all dynamic 2D polymerization, the extent of which depends greatly on the polymerization conditions and polymer system studied“. ¹ We agree with that framework. It is therefore crucial to uncover the basic elements that are involved in the seemingly simple COF synthesis to enable researchers with tool to control further the reaction and products. Further research needs to be conducted, where especially operando methods are critical that can provide a holistic picture of the spatial and temporal landscape of these processes.

Further, we added in the introduction of the manuscript an important sentence referring to Section 1:

However, the emergence of long-range order is a non-trivial process and the subject of recent studies (for a comprehensive overview of the current discussion see Supplementary Information section 1).

And:

At this stage the formed framework is, at least partly, still defective (see also discussion in Supplementary Information section 1).¹⁷

As well as included a new sentence and the suggested reference from Bredas et al.:

To simplify the discussion on the COF assembly process, we describe nucleation, growth and healing as distinct stages, however we note that the crystallization process is complex and these stages can occur at similar timescales.⁴⁴

Also, we changed the below-mentioned sentence in the original text:

This strongly supports the claim that the effect of solvent structuring mainly impacts the rate of the initial, rapid polymerization stage, in accordance with SFME literature.⁵³

B.3 As brief examples, Zhao and coworkers have recently obtained large crystallites using supercritical CO₂ as a solvent (Nat. Commun. 2022). This work certainly does not have phase separation occurring, which the current report discloses as a requirement.

We thank the Reviewer for pointing out this reference. Firstly, we would like to refer to **A.1**: we found structured solvents as a preceding step in conventional solvothermal COF synthesis employing ternary solvent system. As such, the special case of synthesis in supercritical CO₂ is not included in the concept we propose. However, in this paper, Zhao and coworkers raise also important comments about the shortcomings of conventional solvothermal synthesis.

In the paper, the authors spot that in the conventional synthesis method, the spatial proximity of byproducts to the precursors (which they attributed to diffusion processes) can be a source for introducing structural faults into the framework which are counterproductive for the formation of ordered frameworks. Using iSCAT we open a window into understanding by visualizing the way the crystallization process unfolds for the first time. By using ternary solvent systems, structured solvents in the form of surfactant-free (micro)emulsions are unknowingly installed. Thereby, a spontaneous compartmentation of the monomers and catalyst in two distinct phases occurs (see Extended Figure 5). This has, on one hand, the beneficial effect of a kinetic control of the reaction, which is advantageous for the formation an ordered polymer. But, on the other hand, in the case of o/w-emulsions it results in spatial confinement of the monomers and the byproducts in a non-continuous phase which can be detrimental to the formation of ordered polymers. As such, the comments from Zhao and coworkers, resonate well with our description of the impact of structured solvents in conventional solvothermal (imine) COF synthesis.

B.4 Additionally, the amorphous-to-crystalline transformation that the authors rely on has recently been challenged by its original proposers (Dichtel and coworkers), which discloses a significantly more complex pathway than what was previously assumed (J. Am. Chem. Soc. 2022).

We thank the Reviewer for raising this important point. We conducted a survey of previous work to provide an overview of the current picture and the new insights obtained in recent studies and added this discussion to the manuscript and SI (see **B.2**).

We want to emphasize, that regardless of the exact mechanistic pathway of the COF crystallization, the preceding steps of solidification are crucial to understand and control to ensure the optimal conditions for the emergence of a crystalline COF at the end of the process. Our findings using iSCAT reveal that the solidification process is highly advanced and almost completed within the initial 30-60 seconds, a timeframe which hasn't been accessible before, emphasizing the importance of establishing a favorable reaction environment within this preliminary timeframe of the reaction.

Additionally, our new in situ XRD measurements (discussed in part **C** of Reviewer 2) and operando iSCAT observations (e.g. Figure 2) align with literature discussions on a slower reorganization period after the initial solidification, evident in signal plateaus. The duration of this reorganization may vary for different systems, potentially occurring rapidly in certain cases.

B.5 These efforts also do not fully appreciate the computational insight provided to general COF syntheses (Bredas and coworkers, J. Am. Chem. Soc. 2020, 142, 3, 1367–1374,).

We agree with the Reviewer about highlighting also the theoretical efforts to understand the mechanism of COFs synthesis. In our view, the fact that theoreticians assist experimentalists to unlock the crystallization enigma is attesting to the complexity of the process. Here, we present a set of experimental data visualizing the entire process with unprecedented temporal resolution with a greater emphasis on the very first steps of the process. However, with iSCAT we clearly cannot account for the exact way that crystallinity unfolds, at this point. We therefore wish to refrain from speculative discussion and restrict it to the chemistry of the early time frames observed and the correlation with the quality of the resulting product. Please see in **B.2** the changes incorporated to the manuscript as well as the supplementary information, including recent references from Bredas et al.

B.6 Moreover, it is not clear to me how these findings align with formation dynamic studies of metal-organic frameworks, which have also been studied in-depth.

[Redacted text]

[Redacted text]

B.7 As a final note, I mention that the multi-phase nature of COF syntheses was noted in Yaghi's original 2005 report to be important for obtaining crystalline materials. They attribute this to the slowed solubility and kinetics of monomer addition, which seems to be a similar argument to the one made here.

We agree with the Reviewer that throughout the last 19 years there have been several hints that a kinetic barrier needs to be in place for a controlled polymerization and high-quality COFs. While in Yaghi's original report in 2005, the kinetic barrier is installed through a solid and a liquid phase (i.e. the monomers dissolve slowly into the solution and control the reaction), in several other papers, the polymerization dynamics have been on purpose modulated – e.g. with addition of surfactants, chemical modifications of the monomers or by introducing modulators.²

While the beneficial effect of a kinetic barrier has been acknowledged by the community,²⁻⁵ this thought was never extended to the degree that also **in the conventional, solvothermal synthesis** this kinetic control could (and should) already exist. That this could go unnoticed was mainly due to the lack of the required in operando instrumentation (such as iSCAT) that can trace the synthesis proceedings. The use of such methodologies enabled the realization that there is already a kinetic barrier in place that played an important role in the success story of the synthesis of imine based COFs.

Indeed, we agree with the way in which Reviewer #3 formulated this point: "It's the sort of result where in retrospect we should have realized that there should be a connection to surfactant-free emulsions"

B.8 I believe that to understand the insights provided by this report, they must be better contextualized with what has already been done in the field. Other mechanistic reports should also be discussed.

We thank the Reviewer for pointing this out. We included this in the comments above. We think we have now done a very honest description of the state-of-the-art of COFs synthesis and crystallization and we expect that this will assist the readers to understand the relevance of our findings.

B.9 Adding salt to modulate the phases of the liquid solution is interesting and novel.

We thank the Reviewer for pointing to one of the important novelty aspects of our approach. We would like to emphasize here that the use of salt is the outcome of the iSCAT observation obtained in

operando that revealed solvent structuring as a spontaneous basic process. Using the knowledge obtained, we postulate that inverting the phases, namely o/w to w/o, would enable a better reaction landscape in terms of reactant and catalyst compartmentation. We wish to avoid creating the false image of “put some salt and it will make things better” - our approach is broader than that. With the new data included in the revised manuscript (shown in **E**) we undoubtedly expanded the paradigm of modulating the solvent structuring for improving the synthesis of COFs at room temperature by manipulating the reaction environment using generic ternary solvent phase diagrams. Furthermore, in this revision we have a series of new examples and images clearly showing the role of salt addition. They are discussed and shown in more detail in **E**.

C. Data & methodology: validity of approach, quality of data, quality of presentation

C.1 The mechanistic insight provided by iSCAT is interesting. However, the inability to probe the nanostructure of the COFs limits some of the insight provided here. I know the authors collect ex situ PXRD – but I wonder if it would be possible to do in situ PXRD in a capillary condition. I am familiar with the challenges of these measurements, but I think they would greatly augment the findings presented here. It would not be critical to do this on every set of conditions, just the optimized ones (and perhaps unoptimized for comparison). This would remove some of the ambiguities that can be introduced during a work-up step.

We thank the Reviewer for this suggestion. We have performed now in-situ XRDs, shown in Supplementary Figures 49-52 and associated discussion in the manuscript. Our results are very clear and show the emergence of crystallinity at the very early stages when adding salt (13 minutes) compared to the standard conditions without salt (120 minutes) at room temperature. However, before discussing the new results in detail, let us stress some important points.

First, we would like to reinforce that our imaging insights are not testifying to the state of crystallinity. However, we conceptually connect the initial reaction medium condition and embryonic polymer stages with the end product via ex situ measurements and (now also) in-situ XRD analysis. As such, we focus in our manuscript on: the introduction of a novel technique for operando imaging of wet-chemical reactions in material science, the findings of solvent structuring as a preceding step to the framework growth in conventional COF synthesis and finally, the real-life implications of the knowledge obtained by modulating the phases, e.g. by salt addition. The determination of the emergence of crystallinity and the underlying mechanism is therefore at the borderline of the scope of our manuscript. Nevertheless, we acknowledge that the point of first crystalline material in our modified synthesis protocol is of interest to the community, especially in light of the recent discussion by Dichtel, Marder, Evans, Medina, et al. on the synthesis of TA-TAPB COF, as we already discussed in the previous replies to the Reviewer.

Following the Reviewer's recommendation, we have performed in-situ XRD in a capillary for two TA-TAPB COF samples (3M HOAc + 1.5M NaCl, RT and 6M HOAc, RT). By incorporating salt in the synthesis, we register a Bragg reflection of a crystalline COF material related to the 100 plane as early as 13 minutes, whereas without salt and with 6M acetic acid catalyst, detection occurs at ca. 120 minutes. Additionally, in the salt-containing sample, the background notably decreases probably due to a higher number of ordered crystalline particles, enhancing the signal-to-noise ratio, in contrast to the constant high background detected for the 6M sample. Our findings affirm that substituting some of the acetic acid catalyst with salt promotes early development of long-range order, without ambiguity given by the workup procedure. These results support the notion that the formed COF material exhibits fewer structural faults in the initial stages, creating a more favorable platform for the crystallization process. We have now added these results to the main manuscript and Supplementary Information as follows.

Manuscript:

Next, to shed light on the initial emergence of a crystalline product, we conducted in-situ XRD measurements in transmission mode of TA-TAPB COF at RT (see Supplementary Information section 15).

For a 3M HOAc/1.5M NaCl catalyst mixture, a first registration of a reflection assigned to the 100 plane of the COF is evident at the earliest measurement time of 13 min, whereas for the sample containing 6M HOAc catalyst, ordered material can first be detected after ca. 120 min (Supplementary Fig. 50). These observations indicate that through ion-assisted reaction and modulating the solvent structure a framework comprising fewer structural faults has been formed in the early stages of the reaction.

Supplementary Information:

Section 15. In-situ XRD analysis of TA-TAPB COF reaction with and without NaCl

6M HOAc, RT

3M HOAc + 1.5M NaCl, RT

1 h

4 h

8 h

12 h

24 h

Supplementary Figure 49. In situ GIWAXS/SAXS 2D snapshot patterns of TA-TAPB COF synthesized in 1,4-dioxane/mesitylene (v/v 9:1) employing as aqueous catalyst (1 equiv.) either 6M HOAc (left) or 3M HOAc /1.5 M NaCl (right) at RT. Each image was recorded in the time frame of 1h.

Supplementary Figure 50. Data reduction plots of XRD snapshots of the in situ GIWAXS/SAXS 2D analysis obtained for TA-TAPB COF synthesized with 6M HOAc and 3M HOAc / 1.5 M NaCl at different reaction time periods (all: 1,4-dioxane/mesitylene/aqueous catalyst mixture 9:1:1; RT). Each of the diffractograms was recorded in the time frame of 10 min. 3 min preparation time have to be added (see Methods).

Supplementary Figure 51. Data reduction plots of XRD snapshots of the in situ GIWAXS/SAXS 2D analysis obtained for TA-TAPB COF synthesized with 6M HOAc at different reaction time periods (all: 1,4-dioxane/mesitylene/aqueous catalyst mixture 9:1:1; RT). Each of the diffractograms was recorded in the time frame of 1 h.

Supplementary Figure 52. Data reduction plots of XRD snapshots of the in situ GIWAXS/SAXS 2D analysis obtained for TA-TAPB COF synthesized with 3M HOAc / 1.5M NaCl at different reaction time periods (all: 1,4-dioxane/mesitylene/aqueous catalyst mixture 9:1:1; RT). Each of the diffractograms was recorded in the time frame of 1 h. The increasing signal-to-noise ratio is evident and attributed to the increasing number of crystalline particles in the sample.

C.2 I think that the analysis likely leads the authors to the correct conclusion, but formally, the area of the peak is not a valid method to determine crystallinity. Scherrer broadening should be considered a more accurate measure of crystallinity. This compensates for the different overall amount of material (and differences in the X-ray optics) that might vary between measured samples.

We thank the Reviewer for raising this important point. We agree that finding a measure to illustrate trends in crystallinity in XRD patterns is not trivial and that all methods applied as figures of merits have advantages and shortcomings.

Indeed, there have been several reports using the features of diffraction data obtained to express the quality of the COF crystals. Dichtel, Evans, Lotsch and coworkers reported several methods for representing figures of merit, such as the full-width at half maximum (FWHM) of the most intense reflection in the pattern,⁶ the (100) reflection intensity⁷ or the integrated area under the (100) peak.⁸ We tested all of those methods as figures of merit for visualizing the trend that is clearly observed in the experimental data.

XRD reflection broadening, expressed by the FWHM, can be related to the crystallinity character of the samples and can be indicative of the presence of structural irregularities. It is also being used for the calculation of the crystalline domain size via Scherrer equation. However, the use FWHM informs on a specific crystal direction indicated by the particular reflection of miller planes, offering only part of the

picture. In our case, FWHM is evidently not a good measure for the overall crystallinity trend. In the example below, the three graphs presented on the top-right image have a similar FWHM, obviously not capturing the overall crystallinity of the samples.

On the other hand, the (100) reflection intensity (left graph below) gave a similar trend to the one we obtained and showed in the main text at Fig. 4 (right side to compare):

To describe the overall crystallinity of the samples obtained we turned to the calculation of the area under all crystalline reflections inspired by the well-known crystallinity index - also including the up to five different reflections (see right side).

C.3 Overall, the presentation of the data is high-quality.

We thank the Reviewer for the positive confirmation of the quality of our presentation.

D. Appropriate use of statistics and treatment of uncertainties

I believe the uncertainties are well-treated.

We thank the Reviewer for his assessment.

E. Conclusions: robustness, validity, reliability

I find the use of iSCAT interesting and potentially useful. However, I am not sure that the conclusions of this paper are supported by the data provided. The authors claim that the nanostructured solvent is required for effective COF syntheses. There are examples where this is not true. Moreover, the authors do not obtain large single crystals with these conditions, which other groups are now achieving with regularity. The authors claim that these conditions can be used for any COF. They have not shown that this is true beyond a small number of prototypical systems. Perhaps the authors can expand the number of achievable systems.

We thank the Reviewer for bringing up an important discussion point. Based on this comment we expanded the approach to a total of four COF systems (TA-TAPB, TT-ETTA, WTA, TAPB-DMTA), two solvent mixture systems (benzyl alcohol/mesitylene/aqueous HOAc and 1,4-dioxane/mesitylene/aqueous HOAc) and another type of additive (antagonistic salt). This new body of data strengthened and validated our conclusions further and importantly expanded the scope of the ion-assisted approach. The new experimental data shown below (Supplementary Figures 53 to 59) have been added to the Supplementary information and we also expanded the main manuscript by an additional paragraph.

However, before discussing the new data in detail, we would like to address the comment regarding large single crystals.

In the work presented we targeted the very fundamental questions related to the 'common practice and conventional' COF synthesis, i.e., why specific sets of solvents and catalyst (seemingly random) are used as reaction media, why the synthesis through this avenue is so robust and what are the underlying chemical processes that these reaction parameters provide, if any.

We tend to be slightly at variance with the Reviewer regarding the statement that 2D COF single crystals are readily prepared nowadays. Through conventional 2D COF synthesis intergrown polycrystalline COF powder is by far the common product. According to this, in our work, we did not target nor expect to retrieve single crystals with the ion-assisted approach (particularly not at room temperature) but to set the ground principle for its function. The synthesis of 2D COF crystals with a relatively larger domain size (which are outnumbered by 3D COF crystals, to be specific) often requires additional chemical elements that help to slow down the rapid crystallization process. Often (if not always) reaction modulators are implemented into the conventional reaction solvent mixture and at elevated temperatures. We speculate that ion-assisted reactions will play a central role in unlocking the challenge of synthesis of 2D COFs with larger crystalline domain size.

Here we show the new results on modulating the phase diagram of other COFs systems:

Supplementary Information:

Section 16. Expanding the concept to different COFs, solvent systems, and salt types

a) WTA COF, inorganic salt

Supplementary Figure 53. Synthesis scheme of WTA COF in benzyl alcohol and mesitylene with aqueous HOAc as catalyst.

Supplementary Figure 54. PXRD patterns of WTA COF synthesized in benzyl alcohol / mesitylene with different catalyst mixtures (v/v 9:1:1) at room temperature or 100 °C for 3 days.

b) TAPB-DMTA COF, inorganic salt

Supplementary Figure 55. Synthesis scheme of DMTA-TAPB COF in benzyl alcohol and mesitylene with aqueous HOAc as catalyst.

Supplementary Figure 56. PXRD patterns of TAPB-DMTA COF synthesized in 1,4-dioxane / mesitylene with different catalyst mixtures (v/v 9:1:1) at room temperature or 120 °C for 3 days.

c) WTA COF, antagonistic salt

Supplementary Figure 59. PXRD patterns of WTA COF synthesized in benzyl alcohol / mesitylene with different catalyst mixtures (v/v 10:10:1) at room temperature or 100 °C for 3 days.

We discussed our approach, the measurement results, and our interpretation in an added paragraph in our manuscript:

With the thorough insights obtained for the ion-assisted conversion protocol for TA-TAPB COF, we aimed at expanding the solvent structuring paradigm to various COF structures and solvent systems. For this we chose the synthesis of the previously reported COFs WTA,⁶⁵ TT-ETTA⁶⁶ (both synthesized in, benzyl alcohol/mesitylene/aqueous HOAc) and TAPB-DMTA⁴³ (1,4-dioxane/mesitylene/aqueous HOAc) (see Methods). To maintain the solvent structuring established for the TA-TAPB COF, we employed the exact solvent composition i.e., hydrotrope/hydrophobe/aqueous catalyst 9:1:1 v/v, including NaCl salt in the catalyst mixture and conducted the synthesis at room temperature for 3 days. Importantly, the WTA COF powder obtained with the ion-assisted conversion exhibited comparable crystallinity to the material obtained at elevated temperature with the same solvent mixture (Supplementary Fig. 54). In the case of TAPB-DMTA COF, at room temperature, COF powder with high crystallinity was obtained with increased reflection intensity upon increasing salt concentration up to 1.5M (Supplementary Fig. 56). Interestingly, the very same reaction has been conducted at elevated temperatures, however under these conditions powder did not emerge. Moreover, TT-ETTA COF followed the observed trend of increasing crystallinity upon increasing salt concentration (Supplementary Fig. 58)

Notably, the optimal salt amount was found to vary for the different COF systems. This can be attributed to the specific combinations of monomers and solvents, where alterations from the system established mandatorily constitute a change in the shape of the respective phase diagram regions. To probe the influence of employing more hydrophobic solvent mixtures, we turned to yet another very typical

solvent composition namely the previously reported 10:10:1 for WTA COF⁶⁵. Here, by the addition of the aqueous catalyst solution a biphasic separation can coexist along with a structured regime (see also schematically Supplementary Fig. 26^{52,67,68}). The catalyst is thereby mainly confined in a relatively large volume visible to the naked eye. We postulate that breaking the large catalyst confinement droplet into smaller solvent aggregates will serve the polymerization process at room temperature both in terms of reaction medium homogeneity and in the optimum number on nucleation points. Therefore, we aimed at decreasing the water-oil surface tension, thereby decreasing the biphasic regime in the phase diagram, and giving rise to the compartmentation of the catalyst in smaller solvent aggregates. One way of achieving this is the use of a so-called antagonistic salt, e.g. PPhCl₄. Antagonistic salts consist of an organic group with a small inorganic counterion. The ions accumulate at oil/water interfaces, decreasing the surface tension, but do not show activity at air/water interfaces, therefore differing from surfactants.⁶⁹ Strikingly, incorporating PPhCl₄ into the catalyst mixture instead of an inorganic salt (i.e. NaCl) resulted in a highly crystalline WTA COF powder at room temperature (Supplementary Fig. 59). These experiments lay the groundwork for employing liquid phase diagrams to rationally design COF synthesis (see Supplementary Information section 9 for guiding principles).

Furthermore, we added an additional paragraph to the Supplementary information, providing background and guiding principles towards surfactant-free solvent structuring and the underlying phase diagrams:

Section 9. Surfactant-free solvent structuring and liquid phase diagrams

In the following discussion, we aim at elucidating the intricacies of ternary solvent systems, comprising two immiscible fluids and an amphiphilic solvent, and their corresponding liquid phase behaviour. The liquid phase diagrams of these systems can vary significantly depending on the solvents employed (e.g., the single phase region might occupy 5% or 75% of the phase diagram).^{23,28,29} Besides experimentally determining boundaries as well as areas of the their respective subregions, they can also be theoretically predicted by ab initio calculations.²⁹ While the choice of solvents determines the shape of the respective phase diagram, their relative ratio in the ternary mixtures will determine the position in the phase diagram and consequently the structuring.^{23,30} It might be interesting to mention that all participating solvents statistically contribute to each structured phase, implying the presence of, for instance, water within the oil phase, albeit in minor quantities.^{31,32} Importantly, in case of the biphasic/multiphase regime, even in macroscopically separated phases, both can still exhibit structural features.^{28,33,34} For example, Davis and co-workers examined the phase behavior and interfacial tensions of mixtures of hydrocarbon–brine–short chain alcohol (v/v 1:1:1) in view of salinity and temperature.^{28,35,36} They found that increasing either the salinity or temperature could lead to a change in the patterns of phase behavior, from a two-liquid-phase over a three-liquid-phase to a two-liquid-phase equilibrium, as schematically shown in Supplementary Figure 26. The identified changes in phase behavior patterns for the surfactant-free ternary mixtures paralleled those

observed by Winsor in traditional surfactant-containing systems,^{30,37} and the three types of phase equilibria may align with Winsor I, III, and II systems, respectively.²⁸

Supplementary Figure 26. Patterns of phase behavior of the oil-rich, water-rich, and alcohol-rich, microemulsion (ME) phases for equal-volume, surfactant-free mixtures of octane, brine (NaCl (aq.)), and n-propanol. Adapted from Ref. 36. The dependences of phase behavior on the salinity and temperature in surfactant-free ternary mixtures were found to be similar to those in surfactant-containing ones.²⁸

Further, these experiments illustrate that multiple parameters besides the solvents have a considerable impact on the phase diagrams.

For example, increasing the temperature extends the monophasic area while reducing the biphasic region.³⁸ Temperature changes can also alter the relative areas of subregions and influence the size of solvent aggregates.³⁹ Moreover, temperature changes can provide reversible control over phase behavior and the formation of nanodomains.³⁸

Further, additives will have a big influence. For example, inorganic salts can cause significant changes in the location of the subregion boundaries.⁴⁰ They influence the interfacial surface tensions in a similar way as they do for systems containing surfactants (i.e., increasing the tension of water-rich phase/oil-rich phase⁴¹ and alcohol-rich phase/water-rich phase³³; decreasing the tension of alcohol-rich phase/oil-rich phase³³). The effect of salt can also be described as „salting out“ effect: the presence of salt reduces the solubility of other components in the water-rich phase.⁴² Exemplary, it has been reported for the system ethanol/octanol/water, that by salting out the hydrotrope (ethanol) from the water-rich phase, the polarity in the water phase is increased, the system is effectively pushed towards the two-phase region and larger oil-rich domains are formed (octanol).⁴² Other additives investigated include antagonistic salts. Here, it was reported for several systems that they decrease the interfacial tension of the oil/water interface⁴¹ and decrease the two-phase region.⁴³

Another factor that can impact the solvent structure can be the mixing procedure of the solvent components (e.g., the order of addition of the components; step-by-step or one-shot addition).^{34,44} Here, it might be mentioned that the kinetics of their formation are typically fast (< 1s), but can be slowed down.⁴⁵

In recent literature, one of the most studied structuring effects have been the thermodynamically stable surfactant-free microemulsions (SFME). The term SFME is only one of a plethora of names which include

detergentless microemulsions, “pre-Ouzo”, micellar-like structural fluctuations, mesoscale solubilization, and ultraflexible microemulsions (UFMEs).³⁸ The thermodynamic stability of SFMEs can be explained by the establishment of an equilibrium between the repulsive hydration force, attributed to solvation effects, acting as a deterrent to the coalescence of water-rich and hydrophobe-rich domains, and entropy, which propels the system towards the formation of smaller domains.⁴⁶ Interestingly, SFMEs and mesoscopic structuring can also form in binary solvent mixtures.⁴⁷ Furthermore, SFMEs have been designed where the structuring can be specifically controlled by external stimuli, e.g. by CO₂ or temperature.^{39,48} Solvent aggregate sizes exhibit wide variations based on formulation, mixture type, and external parameters, ranging from approximately 1 nm³⁸ to over 100 nm^{23,44} Here, it has been shown that multi-scale aggregates can coexist within the system.^{23,34,38} The role of SFMEs in chemical reactivity is complex, with reports indicating substantial impacts on kinetics, yields, and local reactant concentrations.^{47,49} The complexity increases as the reactants themselves influence the system and phase diagram.

Furthermore, we note that there are several related concepts based on surfactant-free solvent structuring (covering pre-micellar aggregates to microscale entities), such as solvent shifting/displacement,⁵⁰ nanoprecipitation⁴⁴ or facilitated hydrotropy⁵¹.

Furthermore, we performed some additional mechanistical iSCAT studies on the compartmentation effects of structured solvents and the effect of salt on the ternary solvent systems. We added these experiments to the Supplementary Information (Supplementary Information Figures 36 – 40; Supplementary Video 10) and added Extended Figure 5 and some sentences in the main manuscript:

Main:

To probe the compartmentation of the reactants in different solvent regimes, we created a model solvent system (5-fold higher concentration of water), exhibiting stable water-rich droplets on the coverglass surface surrounded by a continuous oil-rich phase. Here, we trace with iSCAT the TA-TAPB COF reaction, which evidently initiates at the oil-water interface and the emergence of the products mainly in the continuous oil phase, both indicating the compartmentation of reactants and catalyst (Extended Data Fig. 5; Supplementary Video 10).

Extended Data Fig. 5. iSCAT imaging of a TA-TAPB COF reaction at the oil-water interface in a model system. Here, aqueous 0.06M HOAc (5 equiv.) was added to 1,4-dioxane/mesitylene (v/v 9:1; 200 μ l) including the reactants TA (2.42 mg, 0.018 mmol) and TAPB (4.2 mg, 0.012 mmol). **a** Upon catalyst addition, stable, water-rich droplets accumulated on the surface (white contrast region). The droplets are surrounded by a continuous oil-rich phase (black contrast region). The low catalyst concentration resulted in a slow reaction and long induction period (hours). The iSCAT image in a) is taken at a point in time, where already initial reactions have taken place, visible in the high contrast white spots in the oil-rich phase from nucleated solid. **b** For these iSCAT images, image a) is subtracted as background (vs. using the initial state before aqueous catalyst addition as background) to show the progression of the iSCAT signal from this point in time. Increasing iSCAT signal fluctuation are detected at the interface of oil- and water-rich phase resulting from the emergence and nucleation of solid phase from a reaction at the interface (see also Supplementary Video 10). The contrast has been adjusted to 0.38 – 1.91. Scalebar (applies to all images), 4 μ m. **c** 3-dimensional surface plot of the iSCAT image after 30 sec (using a) as reference and as background image). The signal stemming from nucleated solid mainly is located in the oil-rich phase, pointing to the compartmentation of catalyst (water-rich) and reactants (oil-rich).

Main:

To confirm the salting-out effect of NaCl on the solvent phases in the TA-TAPB COF system, we conducted iSCAT measurements without HOAc to prevent initiating the reaction (see Supplementary Fig. 36 - 40). While addition of water to the reactant solution resulted in a stable ternary solvent mixture, upon aqueous NaCl addition, water-rich droplets break out of the solution after several minutes and attach on the surface. After 15 min water-rich domains have accumulated on the surface with smaller diameter upon higher salt concentration which we attribute to the increase in surface tension. This shows the role of increasing ionic concentration in structuring the solvent mix.

Supplementary Information Section 14e. Impact of NaCl on rearrangements and solvent system visualized by iSCAT

Supplementary Figure 36. Background-subtracted iSCAT images of the solvent rearrangements in the TA-TAPB reactant mixture upon addition of water (time period: 135 sec). To the initial reactant solution (i ; TA and TAPB in 1,4-dioxane/mesitylene v/v 9:1), water (1 equiv.) is added (ii). Subsequently, phase rearrangement processes and nucleation of black contrast mesitylene droplets (iii) are imaged. The droplets dissolve into solution and a ternary solvent mixture is formed (iv). Images were acquired at a speed of 2.7 ms per frame (371 fps), background-subtracted and 2x2 binned. To enhance visibility, a subset of images was selected from the 50,000 acquired images. Specifically, every 500th image was chosen for display, resulting in a time difference of 1.35 sec between the displayed frames. The contrast is adjusted to 0.71 – 2.08. Scale bar (applies to all images), 4 μm.

Supplementary Figure 37. Background-subtracted iSCAT images of the solvent rearrangements in the TA-TAPB reactant mixture upon addition of aqueous 1.5M NaCl (time period: 110 sec). To the initial reactant solution (i ; TA and TAPB in 1,4-dioxane/mesitylene v/v 9:1), where some undissolved reactants are visible as out-of-focus PSFs, 1.5M NaCl (1 equiv.) is added (ii). In the subsequent phase rearrangement processes, dark-contrast mesitylene droplets nucleate in the water-rich, high-contrast environment (ii). After the dissolution of both phases into solution, white-contrast entities are imaged on the surface for a limited time (iii; see in greater detail Supplementary Figure 38). Finally, the images show a ternary solvent mixture where floating entities exist in solution (iv). Images were acquired at a speed of 2.2 ms per frame (456 fps), background-subtracted and 2x2 binned. To enhance visibility, a subset of images was selected from the 50,000 acquired images. Specifically, every 500th image was chosen for display, resulting in a time difference of 1.1 sec between the displayed frames. The contrast is adjusted to 0.68 – 1.74. Scale bar (applies to all images), 4 μm.

Supplementary Figure 38. Background-subtracted iSCAT images of the solvent rearrangements in the TA-TAPB reactant mixture upon addition of aqueous 1.5M NaCl (time period: 110 ms). Here, the dissolution of the water-rich, white-contrast phase and the dark-contrast mesitylene droplets is shown in higher temporal resolution (transition from ii to iii in Supplementary Figure 37). Images were acquired at a speed of 2.2 ms per frame (456 fps), background-subtracted and 2x2 binned. The contrast is adjusted to 0.5 – 2.0. Scale bar (applies to all images), 4 μm .

Supplementary Figure 39. Background-subtracted iSCAT images taken 3 minutes after adding aqueous 3M NaCl (1 equiv.) to the TA-TAPB reactant mixture (TA and TAPB in 1,4-dioxane/mesitylene v/v 9:1) (time period: 110 sec). In the ternary solvent system formed previously by addition of aqueous NaCl (i), water-rich droplets have nucleated in solution. They are visible in the out-of-focus PSFs (some examples in the initial images have been marked with * for guidance), which attach on the surface during the measurement and give rise to white-contrast, water-rich droplets (ii). Images were acquired at a speed of 13.7 ms per frame (73 fps), background-subtracted and 2x2 binned. To enhance visibility, a subset of images was selected from the 8,000 acquired images. Specifically, every 80th image was chosen for display, resulting in a time difference of 1.1 sec between the displayed frames. The contrast is adjusted to 0.79 – 1.63. Scale bar (applies to all images), 4 μm .

Supplementary Figure 40. To the reaction solution (TA/TAPB in 1,4-dioxane/mesitylene v/v 9:1) water or aqueous NaCl in different concentrations (1 equiv.) were added. Here, the background corrected iSCAT images show the coverglass surface after 15 min. In case of the NaCl mixtures, stable water-rich droplets have nucleated in solution and attached to the surface (see also Supplementary Figure 39). The size of the droplets inversely correlates with the concentration of NaCl, which we attribute to the increasing surface tension of the water-rich phase with increasing NaCl. Scale bar (applies to all images), 4 μm . Contrast is adjusted to 0.66 - 3.12.

F. References: appropriate credit to previous work?

Partially. See above.

We have now incorporated most of the suggested references and enlarged the discussion about the state-of-the-art in synthesis mechanisms for COFs as well early stages of crystallization.

Referee #3 (Remarks to the Author):

It is so rare to have the opportunity to review a manuscript in which the authors demonstrate a true advance that blends fundamental measurements and derived physical chemistry insight with actionable design principles to optimize synthetic procedures, but this manuscript does exactly that. In “Operando reaction imaging demystifies covalent organic framework formation” Gruber and colleagues employ iSCAT imaging that is sensitive to refractive index differences in a solution to image the TA-TAPB COF formation reaction steps of pre-nucleation, nucleation and growth, critically revealing the formation and evolution of emulsion compartments that slow the kinetics of the reaction by limiting the catalyst availability. It’s the sort of result where in retrospect we should have realized that there should be a connection to surfactant-free emulsions, but the authors are providing a big service to the community for not only planting this idea but also for generalizing it in a profound way: the imaging tells a lot already, but they proceed to show a series of ‘salting-out’ control experiments that leverage the ternary solvent diagram in question that ultimately lead to not only an optimization of the amount of acetic acid catalyst needed but to demonstrated predictive power to generalize the principle with another COF and to completely eliminate the need for 3 days of thermal annealing, allowing the annealing to form highly crystalline COFs strictly at room temperature! It is also very compelling to see that the novel iSCAT approach is properly correlated to the outcomes of standard post-anneal characterization methods. This really firmly introduces iSCAT as a tool to advance materials research.

We wish to express our gratitude for the reviewer's very positive evaluation of the work. This paragraph captures in a very explicit fashion the essence of our approach, the context in which this work was done and its future implications.

In my view, this manuscript is a landmark work that is highly worthy of publication in Nature. They have really taken the scientific method full circle, which is refreshing, and this has been a resoundingly successful multidisciplinary collaboration. Furthermore, the work is very carefully done and presented, with very strong experimental backing to the scientific rationale via a comprehensive set of control experiments.

We thank the Reviewer for the very positive note on the novelty and impact of our work and the recommendation of its publication in Nature. Further, we want to thank the Reviewer for highlighting a part of this work that we are especially proud of – connecting nanophotonics and material science in a profound way, resulting in direct implications.

More minor comments:

1. It would help to elaborate in the main text about how the authors know the sizes of the objects they refer to seeing in their movies e.g. 50 nm is mentioned, but the imaging is diffraction limited (might be

wise to cite original iSCAT studies explaining contrast changes as a function of size on p6 line 116?), and also how they know they're seeing something floating in the solution vs on the surface of the coverslip.

We thank the Reviewer for raising this point. We added some references at the indicated sentence (now p7 line 130) and included in the main text the following sentences in order to facilitate the readability and understanding for people not acquainted with iSCAT:

The spatial resolution of an optical technique such as iSCAT is diffraction-limited.³⁵ Therefore, in iSCAT, the signal of a subwavelength particle is detected on the camera as a Point Spread Function (PSF). The PSF closely resembles a 2D Gaussian function with a full-width at half-maximum (FWHM) of roughly half the wavelength of the incident light (i.e., $\text{FWHM}_{\text{PSF}} \gg d_{\text{scatterer}}$).³ However, in iSCAT, the size of the particle can be retrieved from the contrast (amplitude) of the PSF, which scales with the polarizability and therefore with the volume of the scatterer.²⁷ As such, despite being an optical microscope with diffraction-limited resolution, e.g., the signal of a 2 nm gold particle can be detected as a PSF in the camera and by analyzing its contrast the size can be determined.³⁶ Typically, the contrast-size relationship is calibrated by measuring samples of known sizes²⁷ or through theoretical simulations/calculations.^{37,38}

Concerning the size of the visible objects, we specified this with a comment in the Supplementary Information:

Section 2f. Size estimation based on iSCAT contrast

Most of the initial precipitates breaking out of solution and attaching on the coverglass exhibit an iSCAT contrast of around 35% (see Fig 2b v, vi; see also Extended Figure 4).

Supplementary Figure 3. Cross section of an exemplary particle detected in the initial stages. Scale bar, 1 μm .

The iSCAT contrast is proportional to the scattered light field which in turn is proportional to the complex polarizability α of the scatterer. Therefore, the contrast is determined by the volume and the refractive index of the scattering matter and its surrounding medium (see Section 2a). For polystyrene spheres ($n_{\text{PS}} = \text{ca. } 1.62^{17}$) in water ($n = 1.33$), which is a similar system to ours ($n_{\text{COF}} = \text{ca. } 1.59^{19}$; $n_{\text{diox/mes/water}} = 1.41$ - SI section 8), 35% contrast corresponds to a particle diameter of 85 nm. In the COF system, the refractive index of the medium is slightly closer to that of the particle, resulting in less

scattered light compared to the polystyrene case. However, as the scattered light and therefore the contrast scales with the third power of the particle diameter it is fair to assume that the initial particles are in the range of $d \leq 100$ nm. In future studies, we envision that an extensive contrast to volume relation is established either by measuring colloidal COF particles¹⁹ or by theoretical simulation.^{17,20}

And changed it in the main text accordingly:

Interestingly, aggregates of up to 100 nm in diameter break out of the solution and are detected with iSCAT on the second time scale as black-contrast modulations in the images (Fig. 1c iii, ca. 35% iSCAT contrast; for size estimation see Supplementary Information section 2f).

Further, we thank the Review for making us aware that the distinction between something floating in the solution vs. on the surface of the coverslip might be confusing to people not familiar with iSCAT. We added a discussion in the Supplementary Information and referred to it in the main text.

Main:

For differentiation of attached and floating species during this step, please refer to the Supplementary Information section 2d.

SI:

We distinguish between something floating in solution and attached on the surface of the coverglass based on two factors. First, entities floating in solution change their x-y position while something attached on the coverglass is immobilized with a constant x-y position in our case. Second, the z-position of the nanoscale particle changes/fluctuates when floating in solution while it is fixed when attached on the coverglass. This has implications on the observed signal, as the interferometric point spread function (iPSF) depends on the particle position relative to the coverslip and the focal plane.¹⁷ The iPSF is composed of a central peak (exploited for determining sample size) and rings of particular contrast and diameter – both of which are dependent on the axial position of the particle with respect to the focus of the objective.¹⁸ While the on-surface particles are characterized by high contrast of their central peak (in our case, black contrast dots in image; main Fig 1c, iii), for the strongly out-of-focus floating entities only rings are imaged (main Fig. 1c, i).

Furthermore, what are the requirements/constraints on imaging other reactions in the same way.

We added now in the manuscript the particular and general considerations when imaging other chemical reactions:

Importantly, we also show that the TA-TAPB COF sample system is not interacting with the illumination light via optical absorption during iSCAT monitoring, accounting for a truly non-invasive measurement (Supplementary Fig. 17). Another important aspect for imaging reactions and the resulting products with iSCAT, is that after a certain film thickness is reached, the camera saturates due to the high reflection signal. While this is not relevant for the initial reaction stages, to assess the final growth state, we adjusted the absolute volume of reactant solution (100 μl) to prevent camera saturation. Finally, it has to be noted that the signal contrast in iSCAT relies on refractive index changes rather than chemical specificity, necessitating careful consideration and controls, as reflected throughout the discussion.

2. Fig 1c stages are not numbered but referred to with Roman numerals in the caption and text

We thank the Reviewer for pointing this out and changed the position of the numerals in the figure to clarify it:

3. Fig 2b: consider showing up to 2.2 s rather than only to 95 ms to clarify what happens between these two times

We thank the Reviewer for pointing this out. We were indeed already thinking about this. In these 2.2 sec, the solvent structuring is passing through several stages, where mesitylene droplets nucleate, grow, and dissolve. Please, see the corresponding montage that was included in the SI:

Supplementary Figure 23. Background-subtracted iSCAT images of the phase rearrangement processes during TA-TAPB COF formation with 3M HOAc (time period: 1.35 sec). The images show the nucleation, growth, presumably via Ostwald ripening, and dissolution of mesitylene droplets after catalyst addition. Images were acquired at a speed of 2.7 ms per frame (370 fps), background-subtracted, 2x2 binned and temporally averaged (5 consecutive frames). This results in a time difference of 13.5 ms between the displayed frames. The contrast is adjusted to 0.72 – 3.09. Scale bar (applies to all images), 4 μm.

Out of space constraints and for the sake of following the storyline we decided to limit ourselves in Figure 2b to three images that stand exemplary for the addition of aqueous catalyst (Fig. 2b, ii), nucleation of mesitylene droplets (Fig. 2b, iii) and their growth (Fig. 2b, iv), while Fig. 2b, v shows the state after dissolution of mesitylene into solution.

We agree with the Reviewer that we should point this out clearer in the text and added the following in the main text:

While in Fig. 2b only three images (ii – iv) are shown for better readability, they stand exemplary for restructuring processes that take place in 2.2 sec (see Supplementary Fig. 10 and 23).

4. The Presentation of Fig 3b with 'left' and 'right' is a bit awkward. It's hard to understand what is left vs right. Perhaps box left and right separately?

We completely agree. We followed the suggestion and separated the boxes to enhance its clarity. Further, we included now the corresponding iSCAT traces of the induction periods in the Supplementary information.

5. Recheck ref 45.

During the revision process we noticed that the information of the sentence with the reference in question was already included in the introduction and therefore deleted it.

6. I really appreciate the many montages of the movies that are included in the supplement. I wonder if some arrowheads could be added to point to specific key events within these montages to help the reader follow how the movies are interpreted.

We thank the Reviewer for pointing this out. We included several specific key events in the montages to guide the reader, which we agree makes it much easier to follow. Here, we show one exemplary Montage:

Supplementary Figure 21. Background-subtracted iSCAT images of the dissolution of liquid mesitylene droplets (time period: 235 ms). To a model system (i) of 1,4-dioxane/mesitylene (v/v 4:1; 100 μ l) and TA (1.01 mg, 0.045 mmol), water (1 equiv.) was added (ii), leading to the nucleation of dark contrast mesitylene droplets (iii). After their growth via Ostwald ripening, the droplets reach a stable state for a limited period (iv). Subsequently, solvent mixing proceeds, where 1,4-dioxane aids in the dissolution of water throughout the entire volume (v). Consequently, the polarity of the environment decreases, leading to a gradual dissolution of the mesitylene droplets. These interdependent processes manifest in the diminishing white signal surrounding the dark droplet (corresponding to water dissolution) and the reduction in size and decrease in black contrast of the droplet itself until complete dissolution (corresponding to mesitylene dissolution). The resulting ternary solvent system shows an enhanced contrast compared to the initial binary one due to inclusion of water. To enhance visibility, a subset of images was selected from the 50,000 acquired images. Specifically, every 500th image was chosen for display, resulting in a time difference of 2.35 sec between the displayed frames. Images were acquired at a speed of 4.7 ms per frame (212 fps), background-subtracted and 2x2 binned. The contrast is adjusted to 0.43 – 2.12. Scale bar (applies to all images), 1 μ m.

7. Similarly, the extended data figures are extremely helpful to follow the narrative. Consider referring to them/describing them more explicitly rather than only parenthetically, since this will encourage the reader to consult them and enhance the paper's readability.

We thank the Reviewer for this helpful suggestion. In the manuscript, each Figure is now described/referenced explicitly with at least one sentence, while trying to keep the manuscript concise:

Extended figure 1:

Finally, to resolve these processes in higher clarity, without the interference of the reaction, we devised a model system including only one monomer. Here, the nucleation, growth and dissolution of the mesitylene droplets could be observed in greater detail (Extended Data Fig. 1; Supplementary Information section 7).

Extended figure 2:

After the catalyst is distributed in the medium, in-solution polymerization commences, which can be traced with iSCAT by following the fluctuations in the intensity of the detected signal. These intensity oscillations are attributed to the processes of the nucleation and growth of embryonic COF seeds in the reaction medium (see detailed example in Extended Data Fig. 2; Supplementary Video 6).

Extended figure 3:

Strikingly, we are able to follow and visualize the attachment (ca. 3 sec) of a single particle and its subsequent growth (ca. 50 sec) by iSCAT (Extended Data Fig. 3; Supplementary Video 5).

Extended figure 4:

Therefore, the catalyst and the reactants constantly meet each other, which renders the reaction at a high speed and results in a highly defective framework. This formation landscape is schematically visualized in Extended Data Fig. 4a.

Consequently, the contact probability of reactant and catalyst decreases, leading to a reduction of the reaction speed and the extent of kinetic defects formed in the framework (schematically in Extended Data Fig. 4b).

Extended Figure 5:

To probe the compartmentation of the reactants in different solvent regimes, we created a model solvent system (5-fold higher concentration of water), exhibiting stable water-rich droplets on the coverglass surface surrounded by a continuous oil-rich phase. Here, we trace with iSCAT the TA-TAPB COF reaction, which evidently initiates at the oil-water interface and the emergence of the products mainly in the continuous oil phase, both indicating the compartmentation of reactants and catalyst (Extended Data Fig. 5; Supplementary Video 10).

References of the rebuttal letter (blue text)

1. Mähringer, A. *et al.* An electrically conducting three-dimensional iron-catecholate porous framework. *Angew. Chem., Int. Ed.* (2021) doi:10.1002/anie.202102670.
2. Haase, F. & Lotsch, B. V. Solving the COF trilemma: towards crystalline, stable and functional covalent organic frameworks. *Chem. Soc. Rev.* **49**, 8469–8500 (2020).
3. Ji, W. *et al.* Solvothermal depolymerization and recrystallization of imine-linked two-dimensional covalent organic frameworks. *Chem. Sci.* **12**, 16014–16022 (2021).
4. Kang, C. *et al.* Growing single crystals of two-dimensional covalent organic frameworks enabled by intermediate tracing study. *Nat. Commun.* **13**, 1370 (2022).
5. Evans, A. M. *et al.* Two-Dimensional Polymers and Polymerizations. *Chem. Rev.* **122**, 442–564 (2022).
6. Yao, L. *et al.* Covalent Organic Framework Nanoplates Enable Solution-Processed Crystalline Nanofilms for Photoelectrochemical Hydrogen Evolution. *J Am Chem Soc* **144**, 10291–10300 (2022).
7. Feriante, C. *et al.* New Mechanistic Insights into the Formation of Imine-Linked Two-Dimensional Covalent Organic Frameworks. *J. Am. Chem. Soc.* **142**, 18637–18644 (2020).
8. Smith, B. J. *et al.* Colloidal Covalent Organic Frameworks. *ACS Cent. Sci.* **3**, 58–65 (2017).

Reviewer Reports on the First Revision:

Referees' comments:

Referee #2 (Remarks to the Author):

I have read the authors' remarks to my original comments and their response to my co-reviewer (Reviewer 3). I still feel that this work is well-performed and that the additions made by the authors have increased the synthetic scope and characterization quality of this manuscript. I do feel that the findings in this work will be of significance to the COF community.

I respond to each comment individually below. If I omit a comment, I felt that the discussion was resolved.

A1. I sense that the authors are trying to convey that they have unraveled a mechanistic reason for the preparation of single-crystalline 2D COFs in acetic acid / mesitylene / dioxane mixtures. I think their explanation has some validity, based on the iSCAT measurements they show. However, I sense a more obvious reason to use these solvents is because they solvate the monomers effectively. So, I wonder if this solvent ordering is required, or simply assisting the formation of crystalline materials. I note that this is particularly challenging to disentangle because of the challenges associated with work-up, which have been well documented by the authors in previous reports.

I don't think that this (or any) report can unambiguously define that any one feature is required for a general COF synthesis – so I am more just curious how the authors understand their findings in regards to systems where this solvent ordering may not be present?

B2. In the light of the other mechanistic studies that have been performed, can the authors comment on the role of modulators (see *Science*, 2024 383, 1014–101 or *ACS Cent. Sci.* 2019, 5, 11, 18921899, or *J. Am. Chem. Soc.* 2022, 144, 43, 19813–19824) on how this orders solvents needed for successful synthesis? I note that in these syntheses, the reaction times are long and the solvents are totally miscible (in some cases they are single-component, e.g. PhCN) and so presumably the solvent ordering does not occur in these cases?

B6. I think the comment that MOFs polymerize by different mechanisms is somewhat controversial but would be exciting to explore with iSCAT.

C2. I feel that the data as shown in Fig 4b is not meaningful. It could be that the capillary is larger. It could be that the alignment was better. It could be that there are more particles generated. It could be that the X-ray contrast in the presence of salt is better. I highly encourage the authors to not show the data in this way. It is not scientifically accurate.

Instead, I would encourage them to use that space to zoom in on the 2-10 theta region (they could even highlight the area under the curve as a visual cue). But suggesting that the graph of NaCl concentration versus the area under the curve directly indicates the crystal quality is misleading.

I think that the authors discussion about single-crystals and not aiming to isolate them is reasonable. However, this mechanistic insight would have broader appeal if it lead to higher quality materials or enhanced properties related to the higher material quality.

Referee #3 (Remarks to the Author):

The authors have applied considerable additional effort to further generalize their results in their revised manuscript. I appreciate the additional context provided in response to Reviewer 2, in addition to the new experiments and corroborating in situ XRD. I recommend the manuscript be accepted in its current state, modulo some very minor corrections associated with the below list of comments. My only fear is that the manuscript is now so information dense that it will require a very careful read for a general reader to follow (especially the new text regarding the phase behavior, which is highly technical by necessity). I trust that the editor will assist with the communication of the works so that nothing is lost on the reader.

Minor comments:

1. The Roman numerals in Fig 1c are still confusing: the description in the caption does not seem to follow the headings that are in the figure itself, but there are still five of each.

2. In the added manuscript paragraph line 438 "and in the optimum number OF nucleation points."

3. The context in the new supplement Section 1 is very helpful but needs careful editing to ensure proper grammar etc. Also, the sentence "We agree with this framework." unconventionally states an opinion. That should be revised to be more consistent with publishable text rather than a rebuttal.

4. The paragraphing in Section 9 of the supplement could also use some revision, with multiple very brief/single-sentence paragraphs. Specific wording in Section 9: "... from a two-liquid-phase OVER a three-liquid-phase to a two-liquid-phase equilibrium..." what is meant by OVER? Do the authors mean TO?

Author Rebuttals to First Revision:

Point-by-Point Response to Reviewers

Referee #2 (Remarks to the Author):

I have read the authors' remarks to my original comments and their response to my co-reviewer (Reviewer 3). I still feel that this work is well-performed and that the additions made by the authors have increased the synthetic scope and characterization quality of this manuscript. I do feel that the findings in this work will be of significance to the COF community.

First, we would like to thank the Reviewer for the constructive scientific discussion in the presented results and for bringing up critical issues that helped us to sharpen our claims.

I respond to each comment individually below. If I omit a comment, I felt that the discussion was resolved.

A.1 I sense that the authors are trying to convey that they have unraveled a mechanistic reason for the preparation of single-crystalline 2D COFs in acetic acid / mesitylene / dioxane mixtures. I think their explanation has some validity, based on the iSCAT measurements they show. However, I sense a more obvious reason to use these solvents is because they solvate the monomers effectively. So, I wonder if this solvent ordering is required, or simply assisting the formation of crystalline materials. I note that this is particularly challenging to disentangle because of the challenges associated with work-up, which have been well documented by the authors in previous reports.

This is a very important remark made by the Reviewer. The conventional COF synthesis is complex. It can be seen as a one-pot reaction consisting of several components which synergistically contribute to the successful synthesis of a COF. The findings of our work assign to solvents another important role which extends beyond the solubility of the reactants, namely, to being kinetic modulators by compartmentation of reactants and catalyst. With this, we do not underestimate the importance of reactant solubility in the reaction media. It is, naturally, of significant importance and we presume it is an integral part of the solvent structuring installed (e.g. by being capable of altering the surface energy and tension of the different phases). In addition, we assume that non-dissolved precursors will not play a significant role in shaping the characteristics of the solvent structuring. Importantly, with the aid of iSCAT, we can visualize the very first processes occurring upon the addition of an aqueous catalyst into the binary (miscible) precursor-containing solvent mixture solutions. The imaging reveals a rapid phase transformation indicative of a subsequent structuring of the reaction medium. To validate its importance and role in the overall reaction landscape, we conducted several control experiments where solvent structuring is not installed (e.g., binary solvent system of only dioxane/water; addition of mesitylene after the initial stages of reaction). In these experiments, a crystalline TA-TAPB COF at room temperature did not emerge. We, therefore, conclude that under the employed conditions the preliminary barrier is key for obtaining a crystalline COF as end product. It is clearly not the only key process taking place in the overall reaction, but it is among the first critical processes in a multi-stage reaction profile. At this stage, we also cannot comment on the ladder of importance of the occurring processes - we envision the overall COF construction process comprising several complementary elements where solvent structuring is an important part. We don't exclude that in some COF syntheses other processes will have a more critical weight than solvent structuring for the formation of a crystalline COF.

I don't think that this (or any) report can unambiguously define that any one feature is required for a general COF synthesis – so I am more just curious how the authors understand their findings in regards to systems where this solvent ordering may not be present?

See our previous comment regarding our view of the overall conventional COF synthesis. In addition, nowadays, COFs can be made in different ways employing different linkages and different reaction conditions. Generally speaking, the different reaction protocols (non-conventional) will also require to install an appropriate kinetic and thermodynamic balance. When a chemical reaction is very efficient and reversible like the imine and boronic ester bond formation, the synthesis in solvents under elevated temperature will probably require regulation by installing barriers (non-dissolved precursors (solids)/ modulators/ restriction accesses of monomers/time...). We are therefore of the opinion that a reaction that eventually leads to a successful synthesis of a COF mandatorily features a set of checks and balances on the reaction progress. We could show that solvent structuring is an integral element of this toolbox and we believe that in many cases it makes a striking difference.

B.2 In the light of the other mechanistic studies that have been performed, can the authors comment on the role of modulators (see *Science*, 2024 383, 1014–101 or *ACS Cent. Sci.* 2019, 5, 11, 18921899, or *J. Am. Chem. Soc.* 2022, 144, 43, 19813–19824) on how this orders solvents needed for successful synthesis? I note that in these syntheses, the reaction times are long and the solvents are totally miscible (in some cases they are single-component, e.g. PhCN) and so presumably the solvent ordering does not occur in these cases?

[Redacted text]

B.6 I think the comment that MOFs polymerize by different mechanisms is somewhat controversial but would be exciting to explore with iSCAT.

[Redacted text]

[Redacted text]

C.2 I feel that the data as shown in Fig 4b is not meaningful. It could be that the capillary is larger. It could be that the alignment was better. It could be that there are more particles generated. It could be that the X-ray contrast in the presence of salt is better. I highly encourage the authors to not show the data in this way. It is not scientifically accurate.

Instead, I would encourage them to use that space to zoom in on the 2-10 theta region (they could even highlight the area under the curve as a visual cue). But suggesting that the graph of NaCl concentration versus the area under the curve directly indicates the crystal quality is misleading.

We understand the Reviewer's valid concerns. Fig. 4b is designed to illustrate *qualitatively* the trend clearly visible in Fig. 4a. This aims to effectively communicate the information of Fig. 4a in a clearer manner to a broader audience. Considering all figures of merit in hand (presented in the last letter of response), two figures represent this clear trend well; between the two options, namely the peak height (intensity count of the 100 reflection) and the area under the crystalline reflections (background subtracted), we are in favour of the latter because it accounts for the presence of higher order reflections which testify for the quality of the material obtained.

To account for the points made by the Reviewer, we emphasize in the figure caption that Fig. 4b represents a "*qualitative trend of structural order*" seen in Fig. 4a calculated based on the area under all crystalline reflections. We also added the figure of merit using the peak height to the SI:

e) 100 Peak height as Figure of merit for the trend shown in Fig. 4a

Supplementary Figure 36. The plot represents a qualitative trend, visible in Fig. 4a, calculated based on the peak height of the 100 reflection.

I think that the authors discussion about single-crystals and not aiming to isolate them is reasonable. However, this mechanistic insight would have broader appeal if it lead to higher quality materials or enhanced properties related to the higher material quality.

We agree with the Reviewer that higher material quality, particularly of 2D COF, is a highly desired target. We are of the opinion that unravelling the underlining reaction mechanisms and processes of 2D COFs formation under solvothermal conditions is key to enable the synthesis of this type of materials in higher quality, with this mind set we investigated the formation process and wrote our manuscript.

Referee #3 (Remarks to the Author):

The authors have applied considerable additional effort to further generalize their results in their revised manuscript. I appreciate the additional context provided in response to Reviewer 2, in addition to the new experiments and corroborating in situ XRD. I recommend the manuscript be accepted in its current state, modulo some very minor corrections associated with the below list of comments. My only fear is that the manuscript is now so information dense that it will require a very careful read for a general reader to follow (especially the new text regarding the phase behavior, which is highly technical by necessity). I trust that the editor will assist with the communication of the works so that nothing is lost on the reader.

We thank the Reviewer for the important comment. We kept the issue of readability in mind while shortening the text.

Minor comments:

1. The Roman numerals in Fig 1c are still confusing: the description in the caption does not seem to follow the headings that are in the figure itself, but there are still five of each.

We thank the Reviewer to point to this issue the figure and its caption were amended accordingly.

Fig. 1. Interferometric scattering microscopy as operando tool to get holistic insight into the mechanism of COF formation. a Reaction scheme of the TA-TAPB COF formation. The reaction is carried out in a ternary solvent system of 1,4-dioxane, mesitylene and the catalyst mixture, water/HOAc (v/v 9:1:1). **b** Working principle scheme of iSCAT. The incident light is partly reflected at the interface between coverglass and reaction medium. The reflected light interferes with scattered light created by the entities and processes in the reaction mixture and gives rise to the iSCAT signal. The iSCAT image displays the surface of the coverglass and a probe region of around 300 nm into the solution.¹⁷ **c** Top, (i-v) schematic representation of central reaction steps during the formation of a crystalline COF framework.²² Bottom, (i-iv) background-subtracted iSCAT images visualizing spatially and temporally the corresponding dynamic processes. **The images disclose (i) pre-nucleation: the monomers in reactant solution, (ii) catalyst addition: solvent restructuring and nucleation of mesitylene droplets (iii) nucleation and precipitation as well as (iv) growth processes of solid matter during TA-TAPB COF formation (3M HOAc, room temperature);** Scale bar, 4 μm . (v) exemplary PXRD pattern of highly crystalline TA-TAPB COF (6M HOAc, 120 $^{\circ}\text{C}$, 72 h). To obtain highly ordered materials, the framework is typically allowed to find its optimal configuration by reformation of bonds by applying elevated temperature and pressure throughout the synthesis and several days, the solvothermal synthesis (usually 120 $^{\circ}\text{C}$ for 3 days).¹² Powder X-Ray diffraction (PXRD) measurements serve, together with gas adsorption analysis, as probe for the quality of the formed COFs at the end of the process.

2. In the added manuscript paragraph line 438 "and in the optimum number OF nucleation points."

We are grateful for the Reviewer's thorough read; the sentence was changed accordingly as a part of an overall editing of the text.

3. The context in the new supplement Section 1 is very helpful but needs careful editing to ensure proper grammar etc. Also, the sentence "We agree with this framework." unconventionally states an opinion. That should be revised to be more consistent with publishable text rather than a rebuttal.

The suggested phrase has been replaced and the whole text was carefully edited concerning grammar:

Section 1 (now 4). COF crystallization mechanism in current literature

The formation of COFs towards crystalline and porous materials, is a complex process to trace. The conventional techniques to elucidate the processes, are lacking the combination of a suitable time resolution, spatial resolution and sensitivity to the diverse nano-matter present in the mixture, especially at the early stages of molecular interactions and in operando.¹² Because of these limitations, there are still open questions and vibrant discussions around the topic of the exact crystallization mechanism behind the conventional COF synthesis under solvothermal conditions. To tackle the intricacies of these complex systems, even theoretical predictions try to assist with the understanding of the process by performing computational studies on the COF synthesis.^{13,14} In the following, we bring forward a few prominent reports dealing with the experimental investigation of COF growth mechanisms; it is not a review of the literature but a compressive discussion.

Initially, it was almost a consensus for (imine) COFs that during the fast-initial polymerization stage (condensation reactions of the monomers) an amorphous polymer is produced followed by the emergence of long-range order through defect healing by dynamic covalent chemistry (DCC) over a time span of days.^{15,16} However, recent reports by Dichtel, Evans, Medina et al. are pointing towards a more complex process for obtaining a crystalline COF product (e.g., TA-TAPB COF). Medina et al. and several other groups showed the product isolation procedures such as vacuum activation and solvent washing are critical for the long-range order of the COFs obtained. Indicating that the overall crystallinity of those structures is dependent on many factors and that relying on the final product only provides with part of the overall crystallization picture.^{1,17,18} Following this, it has been reported by Dichtel et al., after exploiting gentler activation techniques, that the very initially precipitated species are, at least partially formed as a few-layer disorganized crystalline sheets. In situ synchrotron XRD measurements confirmed that crystalline matter can emerge more rapidly than previously anticipated and that at the earliest possible measurement time of 90 sec ordered material exists in the mixture (TA-TAPB COF).¹⁸ In a later report,¹⁹ Zhao et al. describe the crystallization of TA-TAPB COF process by nucleation and growth stages. At first, a crystalline COF phase can develop rapidly from self-templated monomers along with an amorphous phase which is later transferred via a self-healing growth stage to a crystalline phase. Thereby, the overall crystallinity increases mainly due to the conversion of the amorphous phase to a crystalline phase. This slower amorphous to crystalline transformation has been previously reported on several occasions.^{12,18-20} It is considered that the process is facilitated by the DCC of the initial defective material into long-range ordered COFs (which are more stable towards the conventional vacuum activation).

In a review by Evans et al. the authors proposed that, "a combination of mechanisms is active in all

dynamic 2D COF polymerization reactions, the extent of which depends greatly on the polymerization conditions and polymer system studied".¹²

4. The paragraphing in Section 9 of the supplement could also use some revision, with multiple very brief/single-sentence paragraphs. Specific wording in Section 9: "... from a two-liquid-phase OVER a three-liquid-phase to a two-liquid-phase equilibrium..." what is meant by OVER? Do the authors mean TO?

Following the reviewer comment the whole text has been edited:

Section 9. Surfactant-free solvent structuring and liquid phase diagrams

The following discussion aims at elucidating the intricacies of ternary solvent systems, which comprise two immiscible solvents and an amphiphilic solvent, and their corresponding liquid phase behavior.

The liquid phase diagrams of these systems can vary significantly depending on the character of the solvents employed (e.g., the single phase region might occupy 5% or 75% of the phase diagram).^{23,28,29} In addition to experimentally determining boundaries and areas of the their respective subregions, liquid phase diagrams can also be theoretically predicted by ab initio calculations.²⁹ The shape of the respective phase diagram is determined by the choice of solvents, while the position in the phase diagram and consequently the structuring are determined by the relative ratio of solvents in the ternary mixtures.^{23,30} Notably, all participating solvents statistically contribute to each structured phase, implying the presence of, for instance, water within the oil phase, albeit in minor quantities.^{31,32}

Importantly, in the case of the biphasic/multiphase regime, even in macroscopically separated phases, both phases can still exhibit structural features.^{28,33,34} For example, Davis and co-workers examined the phase behavior and interfacial tensions of mixtures of hydrocarbon–brine–short chain alcohol (v/v 1:1:1) regarding salinity and temperature.^{28,35,36} The authors discovered that increasing either the salinity or temperature could result in a change in the patterns of phase behavior, from a two-liquid-phase to a three-liquid-phase and further back to a two-liquid-phase equilibrium, as schematically shown in Supplementary Figure 26. The observed changes in phase behavior patterns for the surfactant-free ternary mixtures were found to be analogous to those previously reported by Winsor et al. in traditional surfactant-containing systems.^{30,37} Furthermore, the three types of phase equilibria may align with Winsor I, III, and II systems, respectively.²⁸

Supplementary Figure 26. Patterns of phase behavior of the oil-rich, water-rich, and alcohol-rich, microemulsion (ME) phases for equal-volume, surfactant-free mixtures of octane, brine (NaCl (aq.)), and *n*-propanol. Adapted from Ref. 36. The dependences of phase behavior on the salinity and temperature in surfactant-free ternary mixtures were found to be similar to those in surfactant-containing ones.²⁸

Moreover, these experiments demonstrate that multiple parameters, in addition to the solvents, have a significant impact on the phase diagrams.

For instance, increasing the temperature extends the monophasic area while reducing the biphasic region.³⁸ Additionally, temperature changes can alter the relative areas of subregions and influence the size of solvent aggregates.³⁹ Furthermore, temperature changes can provide reversible control over phase behavior and the formation of nanodomains.³⁸

Another considerable influence is achieved through the use of additives such as inorganic salts, which can cause significant alterations to the location of the subregion boundaries in the phase diagram.⁴⁰ In the context of surfactant-free emulsions, these additives have a comparable impact on interfacial surface tensions to that observed in systems containing surfactants (i.e. an increase in the tension of water-rich phase/oil-rich phase⁴¹ and alcohol-rich phase/water-rich phase³³ and a decrease in the tension of alcohol-rich phase/oil-rich phase³³). The effect of salt can also be described as the „salting out“ effect: the presence of salt reduces the solubility of other components in the water-rich phase.⁴² For example, it has been reported for the system ethanol/octanol/water, that by salting out the hydrotrope (ethanol) from the water-rich phase, the polarity in the water phase is increased effectively pushing the system towards the two-phase region and the formation of larger oil-rich domains (octanol).⁴² Other additives investigated include antagonistic salts. Here, it was reported that these additives decrease the interfacial tension of the oil/water interface⁴¹ as well as the two-phase region.⁴³

Another factor that can impact the solvent structure is the manner in which the solvent components are mixed (e.g., the order of addition of the components; step-by-step or one-shot addition).^{34,44} It should be noted here that the kinetics of their formation are typically fast (< 1s), but can be slowed down.⁴⁵

In recent literature, one of the most studied structuring effects has been the thermodynamically stable surfactant-free microemulsions (SFME). The term SFME is one of a multitude of names that include detergentless microemulsions, “pre-Ouzo“, micellar-like structural fluctuations, mesoscale solubilization, and ultraflexible microemulsions (UFMEs).³⁸ The thermodynamic stability of SFMEs can be explained by the establishment of an equilibrium between the repulsive hydration force, attributed to solvation effects, acting as a deterrent to the coalescence of water-rich and hydrophobe-rich domains, and entropy, which

propels the system towards the formation of smaller domains.⁴⁶ It is noteworthy that SFMEs and mesoscopic structuring can also form in binary solvent mixtures.⁴⁷ Furthermore, SFMEs have been designed where the structuring can be specifically controlled by external stimuli, e.g., by CO₂ or temperature.^{39,48} In the case of a phase inversion of an o/w emulsion, the o/w domains gradually grow and interconnect, initially forming a bicontinuous structure and subsequently an oil-continuous domain with w/o domains.²⁸ The sizes of solvent aggregates exhibit considerable variability based on formulation, mixture type, and external parameters, ranging from approximately 1 nm³⁸ to over 100 nm^{23,44}. Here, it has been demonstrated that multi-scale aggregates can coexist within the system.^{23,34,38} The role of SFMEs in chemical reactivity is complex, with reports indicating substantial impacts on kinetics, yields, and local reactant concentrations.^{47,49} The complexity increases as the reactants themselves influence the system and phase diagram.

Furthermore, it should be noted that there are several related concepts based on surfactant-free solvent structuring (covering pre-micellar aggregates to microscale entities), such as solvent shifting/displacement,⁵⁰ nanoprecipitation⁴⁴ or facilitated hydrotrophy⁵¹.